# A high-risk retinoblastoma subtype with stemness features, dedifferentiated cone states and neuronal/ganglion cell gene expression

Jing Liu [1,2,3,34], Daniela Ottaviani [1,2,4,34], Meriem Sefta[1,2,34], Céline Desbrousses [1,2], Elodie Chapeaublanc [1,2], Rosario Aschero [5], Nanor Sirab [1,2], Fabiana Lubieniecki[5], Gabriela Lamas[5], Laurie Tonon[6], Catherine Dehainault[7,8], Clément Hua [1,2], Paul Fréneaux[7], Sacha Reichman [9], Narjesse Karboul[1,2], Anne Biton [1,2,10,11,30], Liliana Mirabal-Ortega [12,13,14], Magalie Larcher[12,13,14], Céline Brulard[1,2,31], Sandrine Arrufat[7], André Nicolas[7], Nabila Elarouci[3], Tatiana Popova[15], Fariba Némati [16], Didier Decaudin[16], David Gentien[16], Sylvain Baulande [17], Odette Mariani[7], Florent Dufour [1,2], Sylvain Guibert[18], Céline Vallot [18], Livia Lumbroso-Le Rouic[19], Alexandre Matet [19,20], Laurence Desjardins[19], Guillem Pascual-Pasto[21,22], Mariona Suñol[21,23], Jaume Catala-Mora [21,24], Genoveva Correa Llano[21,22], Jérôme Couturier[7], Emmanuel Barillot [10,11], Paula Schaiquevich [5,25], Marion Gauthier-Villars[7,8,15], Dominique Stoppa-Lyonnet [7,8,20], Lisa Golmard[7,8,15], Claude Houdayer[7,8,15,32], Hervé Brisse [26], Isabelle Bernard-Pierrot [1,2], Eric Letouzé [27,28], Alain Viari[6], Simon Saule[12,13,14], Xavier Sastre-Garau[7,33], François Doz[20,29], Angel M. Carcaboso [21,22], Nathalie Cassoux[19,20], Celio Pouponnot[12,13,14], Olivier Goureau[9], Guillermo Chantada [4,21,22,25,35], Aurélien de Reyniès[3,35], Isabelle Aerts[1,2,29,35] & François Radvanyi [1,2,35 ✉]

Retinoblastoma is the most frequent intraocular malignancy in children, originating from a maturing cone precursor in the developing retina. Little is known on the molecular basis underlying the biological and clinical behavior of this cancer. Here, using multi-omics data, we demonstrate the existence of two retinoblastoma subtypes. Subtype 1, of earlier onset, includes most of the heritable forms. It harbors few genetic alterations other than the initiating *RB1* inactivation and corresponds to differentiated tumors expressing mature cone markers. By contrast, subtype 2 tumors harbor frequent recurrent genetic alterations including *MYCN*-amplification. They express markers of less differentiated cone together with neuronal/ganglion cell markers with marked inter- and intra-tumor heterogeneity. The cone dedifferentiation in subtype 2 is associated with stemness features including low immune and interferon response, E2F and MYC/MYCN activation and a higher propensity for metastasis. The recognition of these two subtypes, one maintaining a cone-differentiated state, and the other, more aggressive, associated with cone dedifferentiation and expression of neuronal markers, opens up important biological and clinical perspectives for retinoblastomas.

A list of author affiliations appears at the end of the paper.

Retinoblastoma is a rare childhood cancer of the developing retina with an incidence rate of about 1 in 17,000 live births[1–3], but is the most frequent pediatric intraocular malignancy. The main therapeutic objective for retinoblastoma is first to save the child's life through early detection, treatment of the ocular tumor, and prevention of metastatic spread. Secondary goals are eye preservation and maximization of visual potential[4]. In low-income countries, retinoblastoma is associated with low patient survival due to delayed diagnosis, poor access to multi-disciplinary retinoblastoma-specific healthcare, and socioeconomic factors. In high-income countries, tumor remission is achieved in more than 95% of cases, however some patients still develop metastases[5]. Metastases can be due to dissemination through the optic nerve into the central nervous system (CNS) and through the sclera to the orbit. Retinoblastoma can also give rise to systemic metastases[6]. Several histopathological features are considered high-risk factors for tumor progression and metastasis[7].

Retinoblastoma is usually initiated by biallelic inactivation of the *RB1* tumor suppressor gene. A minority of non-hereditary retinoblastomas (<2%) are initiated by *MYCN*-amplification without *RB1* inactivation[8]. In most cases, hereditary retinoblastomas are bilateral, whereas non-hereditary cases are always unilateral.

The retina includes six types of neurons (rod and cone photoreceptors, bipolar, amacrine, horizontal, and ganglion cells) and Müller glia, all of which are generated from multipotent retinal progenitor cells[9,10]. Studies in human show that the cell-of-origin of retinoblastoma is a cone precursor[11–15].

Three studies based on gene expression profiling reached conflicting conclusions concerning the possible existence of retinoblastoma molecular subtypes and the retinal cell type-specific markers expressed in retinoblastoma[16–18]. Beyond *RB1*, the only recurrently mutated gene in retinoblastoma (7–13% of cases) is the epigenetic modifier gene *BCOR*[19–21]. Recurrent genomic alterations have been identified: gains and amplifications on 1q, 2p (targeting *MYCN*), and 6p, losses on 13q (targeting *RB1*) and 16q[22–25]. Several studies have reported a positive correlation between high copy-number alterations, age at diagnosis, and other clinical and histopathological variables, including unilaterality, non-hereditary status, and low differentiation[24,26–30]. Despite this wealth of findings, a molecular framework for understanding the biology and clinical behavior of retinoblastoma is lacking.

In this work, we identify two subtypes of retinoblastoma associated with different clinical and pathological features (age at diagnosis, laterality, heredity, and growth pattern) following integrative analysis of the transcriptome, methylome, and DNA copy-number alteration data from a series of 102 retinoblastomas. Further characterization provides evidence for the relevance of these two subtypes for understanding the biology of retinoblastoma, and for clinical management of this disease. Few genetic alterations other than RB1 inactivation are associated with subtype 1 tumors. By contrast, in addition to RB1 inactivation, almost all subtype 2 tumors harbor other recurrent genetic alterations, including *MYCN* amplifications. Consistent with a maturing cone precursor as the cell-of-origin of retinoblastoma, we find that both subtypes express cone markers. We show, by a detailed analysis of cone differentiation including the use of immunohistochemistry, retinal organoids, and single cells, that subtype 2 tumors are less differentiated than subtype 1 tumors and express neuronal/ganglion cell markers with marked inter- and intratumor heterogeneity. This lower cone differentiation in subtype 2 is associated with stemness features, including a higher propensity for metastasis, as shown by a study of an additional series of 112 retinoblastomas, including metastatic tumors.

## Results

### Identification of two retinoblastoma molecular subtypes with distinct clinical and pathological features

We analyzed a series of 102 enucleated retinoblastomas (Supplementary Data 1). To investigate the existence of different retinoblastoma molecular subtypes, we combined three genomic approaches, mRNA expression, DNA methylation, and somatic copy-number alterations (SCNAs) in a subset of 72 of the 102 retinoblastomas. All three datasets were available for 53 of the 72 tumors, and at least two of the three datasets were available for all 72 tumors (Supplementary Data 1). Within each of these three omics datasets, we calculated several partitions of the samples in k clusters (k-partitions), for various values of k, through unsupervised hierarchical clustering, using varying numbers of features and different linkages (see "Methods" section). Then, for each omics and each value of k, we performed a consensus clustering analysis to derive a consensus k-partition. Doing so the transcriptome-based and methylome-based analyses both yielded stable consensus partitions in two clusters, while the SCNA-based analysis yielded a stable consensus partition in five clusters (Fig. 1a, upper panel and Supplementary Fig. 1a). Cluster memberships from each of the three partitions were analyzed by a cluster-of-clusters approach, briefly, a sample co-classification matrix was built and was then subjected to hierarchical clustering using complete linkage. It revealed the convergence of the three partitions around two molecular subtypes gathering 89% (64/72) of the cases (Fig. 1a, middle panel and Supplementary Fig. 1b). Nearest centroid classification attributed to the same subtypes 63 of the 64 classified samples. Moreover, six of the eight unclassified samples could be attributed to a subtype, yielding a final number of 69 classified samples (69/72, 96%): 31 belonging to subtype 1 and 38 to subtype 2 (Fig. 1a lower panel, and Supplementary Fig. 1c, Supplementary Data 1).

To assign to a subtype the 30 remaining tumors of our 102 tumor series, we then established a nine-CpG-based classifier, based on the genome-wide CpG methylation array profiling (see "Methods" section) (Fig. 1b, left panel and Supplementary Data 1). We verified that there was a high concordance in quantifying the level of CpG methylation between DNA methylation arrays and pyrosequencing assays (Fig. 1b, middle panel). This nine-CpG-based classifier attributed seven of the remaining 30 samples to subtype 1, and 20 to subtype 2, while three cases remained unclassified (Fig. 1b, right panel). Altogether the majority of the tumors (96/102, 94%) could be assigned to one of the two subtypes (38 to subtype 1, 58 to subtype 2).

We then compared the clinical and pathological features of these two subtypes (Fig. 1c, Table 1, Supplementary Data 1). Patients with subtype 1 tumors were significantly younger at diagnosis (median age = 11.0 vs 23.9 months; Wilcoxon rank-sum test, $p = 8.9 \times 10^{-11}$). This subtype included 75% of the bilateral ($p = 1.51 \times 10^{-3}$) and 70% of the hereditary cases ($p = 7.68 \times 10^{-4}$). Unexpectedly, among patients with subtype 1 tumors, age at diagnosis did not differ significantly between hereditary forms (median = 10.2 months) and non-hereditary forms (median = 11.2 months) (Wilcoxon rank-sum test, $p = 0.451$). Likewise, there was also no significant difference between the age at diagnosis for hereditary and non-hereditary forms of subtype 2 tumors (median = 19.8 and 24.7 months, respectively, Wilcoxon rank-sum test, $p = 0.320$). Retinoblastomas generally display exophytic growth (into the subretinal space), endophytic growth (towards the vitreous), or, less frequently, a mixed growth pattern (both endophytic and exophytic). Subtype 1 tumors were significantly more likely to be exophytic, whereas most of the subtype 2 tumors were endophytic ($p = 7.33 \times 10^{-4}$). Necrotic areas were more frequently observed in subtype 2 tumors than in subtype 1 tumors

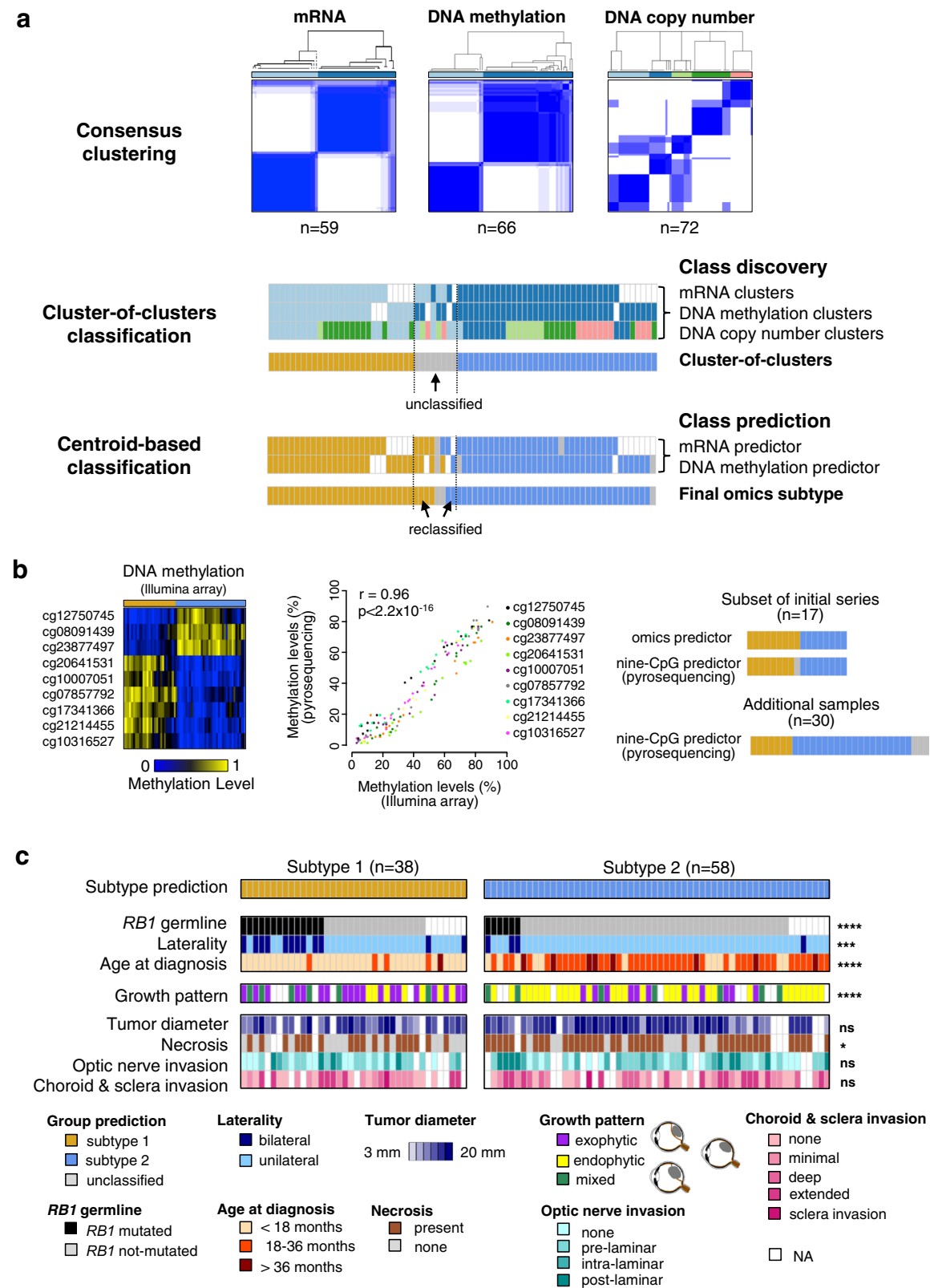

(p = 0.020). Tumor diameter and histological risk features (optic nerve invasion, choroid, or sclera invasion) did not differ significantly between the two subtypes.

**Subtype 2 displayed more genetic alterations than subtype 1 and included the *MYCN*-amplified tumors.** We investigated the genomic characteristics of the two tumor subtypes, by determining their SCNA profiles (Supplementary Data 2). Gains of 1q, 2p (*MYCN*), 6p, 13q, and losses/LOH of 13q (*RB1*), 16q were the most frequent alterations, consistent with reported findings for retinoblastoma[22–25]. 6p gains and 13q losses/LOH were equally distributed between tumor subtypes, whereas 1q gains, 2p gains, and 16q losses/LOH were significantly more frequent in the

**Fig. 1 Multi-omics-based molecular subtypes of retinoblastoma and clinical characteristics. a** Consensus clustering of retinoblastomas based on transcriptomic, DNA methylation, and copy-number alteration data (top panel). Unsupervised cluster-of-clusters analysis (middle panel). Supervised centroid-based classification (bottom panel). Final omics subtype: subtype 1, $n = 31$ (gold); subtype 2, $n = 38$ (blue); unclassified, $n = 3$ (gray). **b** Heatmap showing methylation values (methylome arrays) for the nine-CpG-based classifier (left panel). Correlation between methylation values assessed by pyrosequencing and by methylome array, for 17 tumors (middle panel). A two-sided Pearson's correlation test was used. The nine-CpG-based classifier applied to a subset of 17 tumors of the initial series, led to the same classification as obtained by the -omics approach in 16 cases (one case being not classified by the nine-CpG-based classifier). Subtype assignment of 30 additional tumors based on the nine-CpG-based classifier (right panel). **c** Final molecular classification of 96 retinoblastomas and their key clinical and pathological characteristics. $p \geq 0.05$ (ns), $p < 0.05$ (\*), $p < 0.01$ (\*\*), $p < 0.001$ (\*\*\*), $p < 0.0001$ (\*\*\*\*). For comparisons of *RB1* germline mutation, laterity, growth pattern, tumor diameter, and necrosis between two subtypes, Chi$^2$ tests were used. For comparisons of age at diagnosis and tumor diameter between two subtypes, two-sided Kruskal–Wallis rank tests were used. For comparisons of optic nerve invasion and choroid and sclera invasion between two subtypes, two-sided Fisher's exact tests were used. Exact *p*-values are provided in Table 1.

subtype 2 samples ($p = 5.5 \times 10^{-11}$, $p = 0.0037$, and $p = 1.8 \times 10^{-7}$, respectively) (Fig. 2a). *MYCN* amplifications varied from 14 to 246 copies (Supplementary Data 2) and were found only in subtype 2 tumors (10/58) ($p = 0.013$).

The overall genomic instability score, estimated as the proportion of genome with copy-number alterations, was significantly higher ($p = 3.3 \times 10^{-7}$) for subtype 2 than for subtype 1 tumors (Fig. 2b), and was also significantly higher when tumors with *MYCN* amplification were excluded from the analysis. By contrast, genomic instability scores did not differ between subtype 2 tumors with *MYCN* amplifications and subtype 1 tumors.

We then characterized the mutational landscape of the retinoblastoma subtypes. We performed whole-exome capture followed by paired-end massively parallel sequencing (WES) on genomic tumoral and matched normal DNA of 71 patients from the 102-retinoblastoma series (subtype 1, $n = 25$; subtype 2, $n = 41$; unclassified, $n = 5$). We identified 242 somatic mutations in 186 genes (Supplementary Data 2). The tumors harbored a median of two mutations. The number of somatic mutations identified by WES was significantly higher ($p = 1.2 \times 10^{-7}$) for subtype 2 than for subtype 1 tumors (Fig. 2c). Restricting subtype 2 tumors to either *MYCN*-amplified or *MYCN*-non-amplified tumors yielded the same result.

Three genes, *RB1*, *BCOR*, and *ARID1A*, were found to be recurrently mutated. We performed targeted sequencing for these three genes in 23 of the 31 samples lacking WES data. The distributions of *RB1*, *BCOR*, *ARID1A* mutations, *MYCN* amplifications, 1q gains, and 16q losses are shown by subtype in Fig. 2d. For *RB1* the germinal and somatic point mutations identified are shown, together with deletions, copy-neutral LOH, and promoter methylation. *RB1* mutations were found in most tumors, regardless of subtype, and no difference in the mutation type was observed between the two tumor subtypes. Of note, we found a tumor without *RB1* alteration, it belonged to subtype 2 and displayed a high level of *MYCN* amplification (141 copies). *BCOR* mutations ($n = 9$) were found exclusively in subtype 2 ($p = 0.02$), as were the two *ARID1A* mutations. Most of the subtype 2 tumors without *MYCN* amplification (46/48, 96%) presented gains of 1q and/or losses of 16q. By contrast, none of the *MYCN*-amplified tumors except one had a 1q gain or 16q loss ($p = 0.005$) (Fig. 2d).

**Subtype 2 tumors harbored hypermethylation within CpG islands and hypomethylation outside CpG islands.** We compared the methylome of subtype 1 tumors ($n = 27$) and subtype 2 tumors ($n = 36$, including 4 *MYCN*-amplified tumors). A heatmap representing the methylation levels of the 6607 CpGs significantly differentially methylated between the two subtypes (Supplementary Data 2) is shown in Fig. 2e. Subtype 2 tumors showed more frequent hypermethylation within CpG islands, and

a more frequent hypomethylation outside CpG islands, than subtype 1 tumors (Fig. 2f, g and Supplementary Fig. 2). The four *MYCN*-amplified subtype 2 tumors studied presented a hypomethylation outside CpG islands and did not present hypermethylation within CpG islands (Fig. 2g).

**The two subtypes exhibited differences in the expression of cone and ganglion/neuronal markers and in stemness.** We compared the transcriptome of the two subtypes. Almost one-third of the genes were found differentially expressed between the two subtypes (6207/20408, adjusted *p*-value < 0.05) (Supplementary Data 3).

Cone markers (such as *GUCA1C*, *GNAT2*, *ARR3*, *GUCA1A*, *GUCA1B*, *GNGT2*, *PDE6C*, *PDE6H*, *OPN1SW*) and neuronal/ganglion markers (such as *EBF3*, *DCX*, *ROBO1*, *SOX11*, *GAP43*, *PCDHB10*, *STMN2*, *NEFM*, *POU4F2*, *EBF1*) were among the most differentially expressed genes. Cone markers were over-expressed in subtype 1 tumors, whereas neuronal/ganglion markers were overexpressed in subtype 2 tumors (Fig. 3a). Among the genes known to be involved in retinoblastoma[1,31], several were found to be differentially expressed between the two subtypes (*KIF14*, *MDM4*, *MIR17HG*, *MYCN*, *SKP2* upregulated in subtype 2; *RBL2* downregulated in subtype 2) (Supplementary Data 3). Some of these genes were located in gained/amplified (*KIF14* and *MDM4* at 1q32.1 and *MYCN* at 2p24.3), or lost (*RBL2* at 16q12.2) chromosomal regions, whereas others were involved in the MYC/MYCN pathway (*MIR17HG*, *SKP2*). Hierarchical clustering of the 6207 genes identified three main gene clusters: two upregulated in subtype 1 (gene cluster 1.1 consisting of 1201 genes and gene cluster 1.2 consisting of 1788 genes) and one containing all the genes upregulated in subtype 2 (3112 genes; gene cluster 2) (Fig. 3b). We performed enrichment analysis using the gene sets from gene ontology biological processes (GOBP) and MSigDB hallmarks (HALLMARK) (Fig. 3c and Supplementary Data 3). Cluster 1.1 genes mainly upregulated in a subset of subtype 1 tumors, were associated with tumor microenvironment (immune response, inflammation, interferon response, complement, glial cells) and rod cells markers. Cluster 1.2 was enriched in genes related to fatty acid metabolism, oxidative phosphorylation, and photoreceptor/cone cells. Cluster 2 was enriched in genes associated with the cell cycle, E2F target genes, RNA processing, MYC pathway, and neuron morphogenesis.

The lack of an inflammation/immune signature and the enrichment in MYC and E2F target genes in subtype 2 was evocative of stemness features[32,33]. Moreover, *CD24*, one of the two most overexpressed genes in subtype 2 tumors (Fig. 3a and Supplementary Data 3), has been shown to be a neuronal stem cell marker and a cancer stem cell marker for several tumor types[34]. Stemness indices, based on transcriptomic data, allowed relative evaluation of the degree of stemness in tumor samples. We applied four different stemness signatures[32,33,35,36] to the 59

**Table 1 Clinical and histopathological characteristics of patients stratified by molecular subtype.**

|  | Subtype 1 | Subtype 2 |  |  |
|---|---|---|---|---|
|  | n (%) | n (%) | N | p-value[a] |
| Patients | 38 (40) | 58 (60) | 96 |  |
| Clinical Center |  |  |  |  |
| Institut Curie | 31 (42) | 43 (58) | 74 | 0.655[b] |
| Hospital Garrahan | 6 (33) | 12 (66) | 18 |  |
| Hospital Sant Joan de Déu | 1 (25) | 3 (75) | 4 |  |
| Sex |  |  |  |  |
| Female | 17 (35) | 31 (65) | 48 | 0.403[c] |
| Male | 21 (44) | 27 (56) | 48 |  |
| *RB1* germline mutation |  |  |  |  |
| Yes | 14 (70) | 6 (30) | 20 | $7.681 \times 10^{-4}$ [c] |
| No | 17 (28) | 44 (72) | 61 |  |
| NA | 7 (47) | 8 (53) | 15 |  |
| Laterality |  |  |  |  |
| Bilateral | 12 (75) | 4 (25) | 16 | $1.506 \times 10^{-3}$ [c] |
| Unilateral | 26 (33) | 54 (66) | 80 |  |
| Age at diagnosis |  |  |  |  |
| <18 months | 33 (73) | 12 (27) | 45 | $2.132 \times 10^{-9}$ [d] |
| 18–36 months | 4 (10) | 38 (90) | 42 |  |
| >36 months | 1 (11) | 8 (89) | 9 |  |
| Growth pattern |  |  |  |  |
| Endophytic | 7 (18) | 31 (82) | 38 | $7.332 \times 10^{-4}$ [c] |
| Exophytic | 19 (63) | 11 (37) | 30 |  |
| Mixed | 6 (46) | 7 (54) | 13 |  |
| NA | 6 (40) | 9 (60) | 15 |  |
| Tumor diameter (mm) |  |  |  |  |
| (3.98–6.67] | 1 (50) | 1 (50) | 2 | 0.2094[d] |
| (6.67–9.33] | 1 (25) | 3 (75) | 4 |  |
| (9.33–12] | 7 (50) | 7 (50) | 14 |  |
| (12–14.7] | 9 (64) | 5 (36) | 14 |  |
| (14.7–17.3] | 9 (27) | 24 (73) | 33 |  |
| (17.3–20] | 5 (31) | 11 (69) | 16 |  |
| NA | 6 (46) | 7 (54) | 13 |  |
| Necrosis |  |  |  |  |
| Yes | 18 (31) | 40 (69) | 58 | 0.0203[c] |
| None | 16 (57) | 12 (43) | 28 |  |
| NA | 4 (40) | 6 (60) | 10 |  |
| Optic nerve invasion |  |  |  |  |
| None | 12 (48) | 13 (52) | 25 | 0.7467[b] |
| Prelaminar | 12 (39) | 19 (61) | 31 |  |
| Intralaminar | 4 (33) | 8 (66) | 12 |  |
| Post-laminar | 4 (31) | 9 (69) | 13 |  |
| NA | 6 (40) | 9 (60) | 15 |  |
| Choroid and sclera invasion |  |  |  |  |
| None | 10 (40) | 15 (60) | 25 | 0.6468[b] |
| Minimal | 10 (48) | 11 (52) | 21 |  |
| Deep | 1 (14) | 6 (86) | 7 |  |
| Extended | 8 (38) | 13 (62) | 21 |  |
| Sclera invasion | 1 (50) | 1 (50) | 2 |  |
| NA | 8 (40) | 12 (60) | 20 |  |

*NA* not available, *n* number in each subtype, *N* total number.
[a]Significant *p*-value < 0.05.
[b]Two-sided Fisher's exact test.
[c]Chi$^2$ test.
[d]Two-sided Kruskal–Wallis rank test.

targets, MYC targets V2, MYC targets V1 and G2/M checkpoint (Fig. 3d, lower panel and Supplementary Data 3). These hallmarks were the same as those identified in cluster 2 (cluster of genes overexpressed in subtype 2). The hallmarks negatively correlated with stemness included interferon-alpha response, interferon-gamma response, and complement (Fig. 3d, lower panel and Supplementary Data 3), and were the same as those identified in cluster 1.1 (cluster of genes overexpressed in subtype 1 and associated with the tumor microenvironment). We also assessed the relationship between stemness and the abundance of the various immune cells, as estimated with the Microenvironment Cell Population (MCP)—counter score[37]. Stemness indices were negatively correlated with the MCP scores of monocytic lineage, B lineage, and cytotoxic lymphocytes (Fig. 3d, lower panel and Supplementary Data 3). Altogether, we showed that subtype 2 was associated with high stemness.

The upregulation of cone-related genes in subtype 1 and of neuronal/ganglion cell-related genes in subtype 2 (Fig. 3a) led us to analyze in detail the expression of genes associated with the different retinal cell types (rod and cone photoreceptors, ganglion, amacrine, bipolar, and horizontal cells, and Müller glia). The list of retinal cell type markers was selected from a systematic literature search and from single-cell RNA-seq (scRNA-seq) data obtained at different time points during human retinal development[38]. From the annotated cell types defined by Lu et al.[38], we identified lists of candidate markers associated with each retinal cell type (Supplementary Data 3). In order to choose the most specific markers, we developed a tool for visualizing gene expression profiles in the different retinal cell types (see "Methods" section) (https://retinoblastoma-retina-markers.curie.fr/). Based on an analysis of the expression profiles of the candidate markers obtained from Lu et al.'s data and of markers found in the literature we proposed markers for the different retinal cell types (given in Supplementary Data 3).

Cone markers were overall expressed in both subtype 1 and 2 retinoblastomas, with different expression levels between subtypes depending on the markers (Fig. 3e, upper panel). Among the 24 ganglion cell markers analyzed, a small subset (*EBF3, EBF1, GAP43, POU4F2, NEFM, ALCAM, NRN1, CNTN2*) were consistently overexpressed in subtype 2 tumors (Supplementary Fig. 4a and Fig. 3e, lower panel).

Using the lists of candidate markers associated with each retinal cell type obtained from Lu et al.'s data[38], we provided further evidence for an enrichment of markers associated with ganglion cells in subtype 2 tumors (Supplementary Data 3). These genes overexpressed in subtype 2 tumors can be considered both as ganglion and neuronal genes. Indeed, although specific to ganglion cells in the context of the retina (Supplementary Fig. 4b), all displayed expression in the brain and played different functions in the central nervous system[39–47].

Most of the markers of other retinal cell types (rods, amacrine, bipolar, horizontal, and Müller glia cells) were not expressed in retinoblastomas or were only expressed in a subset of tumors (Supplementary Fig. 4a). The expression of these markers was likely due to the presence of normal retinal cells in some retinoblastomas. Indeed non-neoplastic rods and Müller glial cells have been shown to be present in some retinoblastomas[13].

retinoblastoma samples for which transcriptomic data were available. The stemness indices assessed by these signatures were significantly higher in subtype 2 than in subtype 1 (Fig. 3d, upper panel and Supplementary Fig. 3a). In addition, the stemness indices obtained with the different signatures were highly correlated (Supplementary Fig. 3b). We searched for hallmark gene sets associated with stemness (Supplementary Data 3). The hallmarks positively correlated with stemness included E2F

**State of cone differentiation and expression of neuronal/ganglion cell markers distinguished the two subtypes.** The expression of cone markers observed in both subtypes of retinoblastoma is consistent with the retinoblastoma cell-of-origin being a committed cone cell. Differences in cone marker expression were observed between the two subtypes, raising the question of whether these differences could correspond to

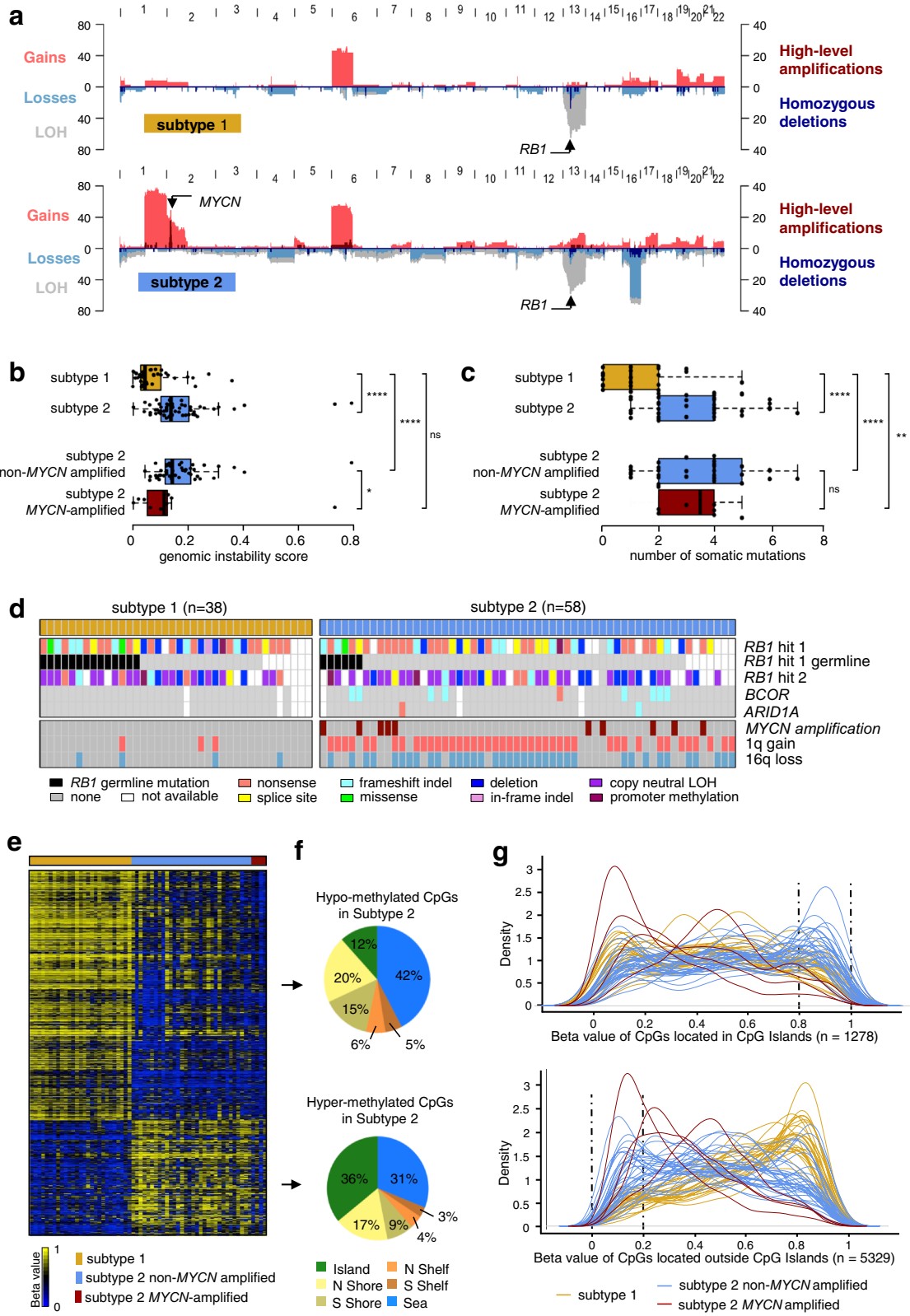

different stages of cone differentiation. Retinal organoids are three-dimensional structures derived from human induced pluripotent stem (iPS) cells that recapitulate the spatial and temporal differentiation of the retina providing powerful in vitro models of human retinal development[48,49]. We measured the level of expression of early and late cone markers in retinal organoids at various time points (d35, d49, d56, d84, d112, d175) after the

differentiation of human iPS cells into the retina, and in subtype 1 (n = 23) and subtype 2 (n = 44) retinoblastomas, using the NanoString technology (Fig. 4a and Supplementary Data 4). As expected, in iPS cell-derived retinal organoids, the expression of early photoreceptor/cone markers (OTX2, CRX, THRB, RXRG) appeared at earlier time points than late cone markers (PDE6H, GNAT2, ARR3, GUCA1C). GUCA1C was the last marker to be

**Fig. 2 Genomic characterization, somatic mutational landscape, and DNA methylation profiles of the two retinoblastoma subtypes. a** Pattern of somatic copy-number alterations in subtype 1 (top, $n = 38$) and subtype 2 (bottom, $n = 58$) retinoblastomas. **b** Boxplots comparing genomic instability between subtype 1 tumors ($n = 38$) and subtype 2 tumors ($n = 58$). Among the subtype 2 tumors, non-MYCN-amplified ($n = 48$) and MYCN-amplified ($n = 10$) tumors are also shown. Significant differences were tested by two-sided Wilcoxon tests for Subtype 1 vs Subtype 2: $p = 3.3 \times 10^{-7}$; Subtype 1 vs Subtype 2 non-MYCN: $p = 1.2 \times 10^{-7}$; Subtype 1 vs Subtype 2 MYCN-amplified: $p = 0.147$; and Subtype 2 non-MYCN-amplified vs Subtype 2 MYCN-amplified: $p = 0.014$. **c** Boxplots comparing the number of somatic mutations between subtype 1 tumors ($n = 25$) and subtype 2 tumors ($n = 41$). Among the subtype 2 tumors, non-MYCN-amplified ($n = 33$) and MYCN-amplified ($n = 8$) tumors are also shown. Significance differences were tested by two-sided Wilcoxon tests for Subtype 1 vs Subtype 2: $p = 8.1 \times 10^{-7}$; Subtype 1 vs Subtype 2 non-MYCN-amplified: $p = 3.5 \times 10^{-6}$; Subtype 1 vs Subtype 2 MYCN-amplified: $p = 0.001$; and Subtype 2 non-MYCN-amplified vs Subtype 2 MYCN-amplified: $p = 0.775$. **b, c** In the boxplots, the central mark indicates the median and the bottom and top edges of the box the 25th and 75th percentiles. The whiskers are the smaller of 1.5 times the interquartile range or the length of the 25th percentiles to the smallest data point or the 75th percentiles to the largest data point. Data points outside the whiskers are outliers. Note: $p \geq 0.05$ (ns), $p < 0.05$ (*), $p < 0.01$ (**), $p < 0.001$ (***), $p < 0.0001$ (****). **d** Somatic mutations of the three genes recurrently altered by tumor subtype. For RB1 are indicated the germline mutations. MYCN amplifications, 1q gains, and 16q losses are also shown. **e** Heatmap of the 6607 differentially methylated CpGs (difference of methylation level >0.2, adjusted $p < 0.05$, two-sided Wilcoxon test and BH correction) between subtype 1 and subtype 2. **f** Distribution, in subtype 2 as compared to subtype 1, of hypomethylated CpGs (upper panel) and hypermethylated CpGs (lower panel), by CpG content and neighborhood context. **g** Density plots showing the distribution of methylation levels of the differentially methylated CpGs located in CpG islands (upper panel) and outside CpG islands (lower panel).

expressed, consistent with previous in vitro and in vivo observations[50,51]. Early cone markers were expressed in both tumor subtypes, at very similar levels. By contrast, late cone markers were expressed, on average, at lower levels in subtype 2 tumors, the most downregulated marker GUCA1C being the latest cone marker expressed. These results indicated that subtype 1 tumors corresponded to a more differentiated stage of cone development than subtype 2 tumors.

As several neuronal/ganglion cell lineage-related genes were shown to be differentially expressed between tumor subtypes (Fig. 3), we also compared their levels of expression in retinal organoids and in tumor samples of the two subtypes (Fig. 4a and Supplementary Data 4). Ganglion-cell markers were expressed at early time points of retinal differentiation (from d49), and their expression levels decreased after d84, consistent with the loss of ganglion cells in retinal organoids at late time points[52]. These ganglion markers were upregulated in subtype 2 compared to subtype 1 tumors (Fig. 4a). Two of them, EBF3 and GAP43, were expressed in subtype 2 tumors with levels comparable to those observed in retinal organoids between d49 and d84.

To more precisely determine the cone development stage corresponding to subtype 1 and subtype 2 tumor cells, we calculated, for each time point after the induction of retinal differentiation, the correlation coefficient between the centroid of each tumor subtype and those of the organoids using cone marker expression (Fig. 4b). Subtype 1 tumors were closest to later cone differentiation (highest correlation observed at d173), whereas subtype 2 tumors were closest to earlier cone differentiation (highest correlation observed between d84 and d112).

To illustrate the degree of cone differentiation achieved by individual retinoblastoma cases of each subtype, we generated a phylogenetic tree using photoreceptor/cone marker expression, incorporating retinal organoid samples at various time points after the induction of differentiation, and retinoblastoma samples (Fig. 4c). All subtype 1 tumors were close to iPS cell-derived retinal organoids at a late time point of differentiation (d173). Subtype 2 tumors were spread out from d84 to d173 retinal organoids.

To explore further the heterogeneity in terms of cone differentiation in retinoblastoma, we studied by immunohistochemistry the distribution of an early photoreceptor marker (CRX), and a later marker specific to the cone lineage (ARR3). We also assessed the expression of one ganglion cell marker (EBF3). Immunohistochemical staining was performed on paraffin-embedded samples of subtype 1 ($n = 9$) and subtype 2

($n = 25$) retinoblastomas (Supplementary Data 4). Two examples of each tumor subtype are presented in Fig. 4d. As expected, in the peritumoral normal retina, the transcription factor CRX was expressed in all cells of the outer nuclear layer (ONL), whereas ARR3 was expressed in a subset of cells in the ONL. EBF3 was expressed in ganglion cells, but also in some amacrine cells in the inner nuclear layer, as previously reported[51,53–55]. All tumors, regardless of the subtype, expressed the photoreceptor marker CRX in agreement with retinoblastoma being derived from cone-committed cells. The ARR3+/EBF3− pattern was the only pattern observed in subtype 1 tumors (Fig. 4d, e and Supplementary Data 4). These tumors were positive for the proliferation marker Ki-67 (Fig. 4d, and Supplementary Data 4). Two types of expression patterns were observed for ARR3 and EBF3 in subtype 2 tumors (Fig. 4d). Most subtype 2 tumors (16/25, 64%) coexpressed ARR3 and EBF3 (ARR3+/EBF3+), as illustrated by tumor RB659 in Fig. 4d. Other subtype 2 tumors (8/25, 32%) displayed mutually exclusive expression of ARR3 and EBF3 (ARR3−/EBF3+ or ARR3+/EBF3− areas), as illustrated by tumor RB617 in Fig. 4d. One tumor (1/25) expressed EBF3 but not ARR3. Tumors of subtype 2 coexpressing ARR3 and EBF3 (ARR3+/EBF3+) were always positive for Ki-67. In subtype 2 tumors with a mutually exclusive expression of ARR3 and EBF3, the ARR3−/EBF3+ areas were always positive for Ki-67, whereas the ARR3+/EBF3− areas were mostly negative for Ki-67 (6 of 7 cases tested) (Fig. 4d and Supplementary Data 4). Histological examination of these Ki-67-negative ARR3+/EBF3− areas showed the presence of fleurettes (foci of photoreceptor differentiation) and an absence of mitoses in three of these six cases. The presence of these different areas within the tumor could reflect a range of tumor cell type stages, from stem, to progenitor to differentiating to terminally differentiated, with many of the latter being post-mitotic. Alternatively, the Ki-67-negative ARR3+/EBF3− areas could correspond to retinoma, a benign non-proliferative lesion observed adjacent to retinoblastoma[56–58].

**Single-cell analysis of intratumoral heterogeneity in a subtype 2 tumor.** To further explore the intratumoral heterogeneity of subtype 2 tumors, we performed droplet-based single-cell RNA sequencing on a subtype 2 tumor (RBSC11). Immunohistochemical analysis of this tumor showed a mutually exclusive expression of ARR3 and EBF3, defining two types of areas (CRX+/ARR3+/EBF3− and CRX+/ARR3−/EBF3+) (Supplementary Fig. 5a), as observed in about 30% of subtype 2 tumors.

We retained transcriptomes of 1198 cells after initial quality controls (Supplementary Fig. 5b). To identify the different cell

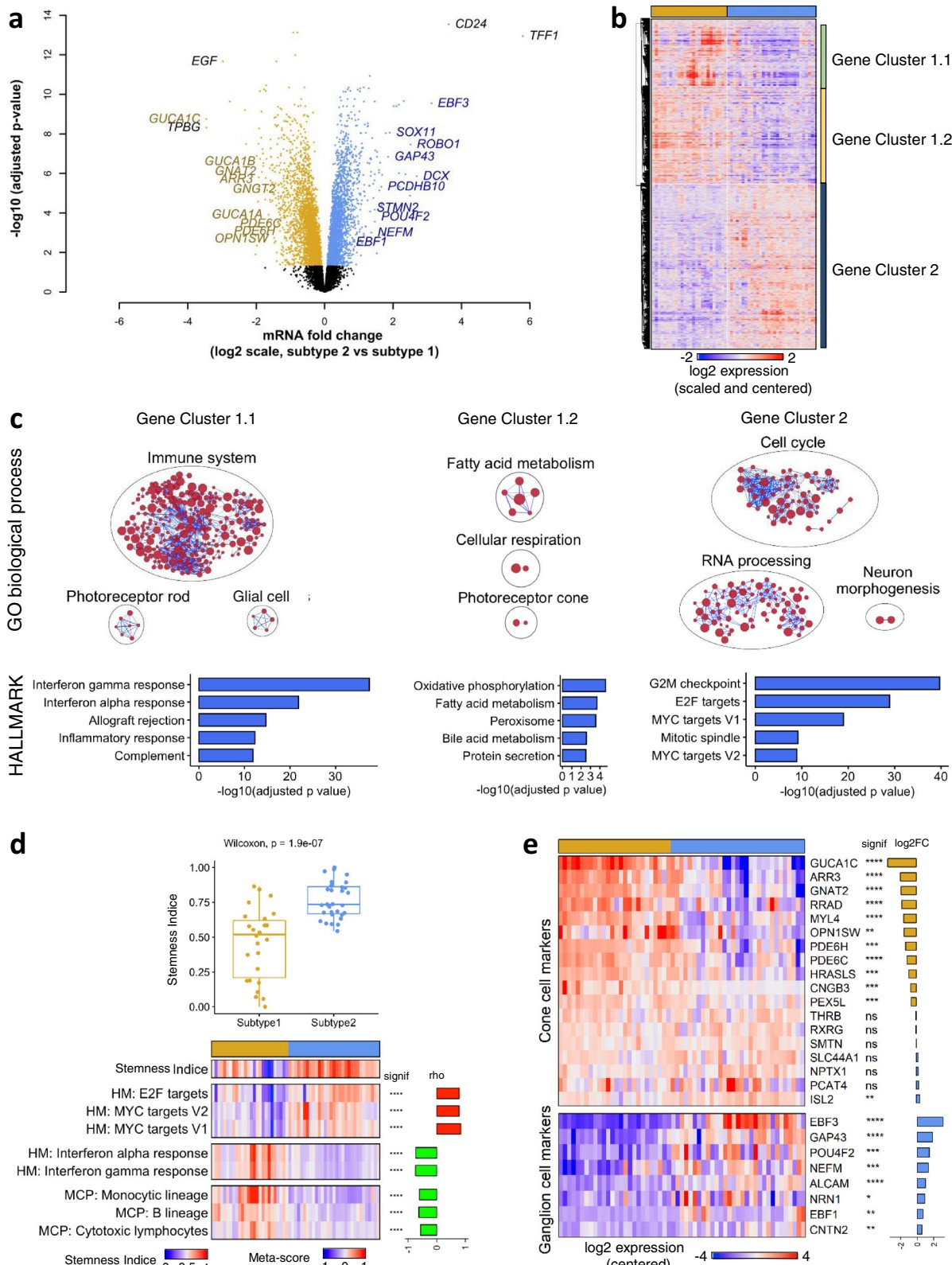

populations present in the tumor, we performed shared nearest neighbor (SNN) clustering and identified seven clusters (Fig. 5a).

To characterize the different clusters, we used (1) known cell type-specific markers, (2) cluster markers (the most upregulated genes in the cluster compared to all other clusters), (3) pathway analysis of cluster markers, (4) correlation to bulk mRNA expression profiles of purified cell types (Fig. 5b, c and

Supplementary Fig. 5c, d, Supplementary Data 5). Clusters 0–4, accounting for 89.2% of all cells analyzed, expressed early photoreceptor/cone markers (e.g., *OTX2*, *CRX*, *THRB*, and *RXRG*). Clusters 0 and 2 expressed neuronal/ganglion cell markers (e.g., *GAP43*, *SOX11*, *UCHL1*, *DCX*, *EBF3*), whereas clusters 1 and 4 expressed late cone markers (e.g., *ARR3* and *GUCA1C*). Clusters 2 and 4 expressed proliferation markers, such

**Fig. 3 Transcriptomic differences between the two retinoblastoma subtypes. a** Volcano plot with genes significantly upregulated in subtype 1 ($n = 26$) (gold) and subtype 2 ($n = 31$) (blue). The genes related to cone-cell and neuronal/ganglion-cell differentiation are indicated (in gold and blue, respectively), together with the most highly differentially expressed genes in each subtype. **b** Hierarchical clustering of the significantly differentially expressed genes identified three main gene clusters. **c** Upper panels: Gene sets from the GOBP collection enriched in clusters 1.1, 1.2, 2 in hypergeometric tests. Results are presented as networks of enriched gene sets (nodes) connected based on their overlapping genes (edges). Node size is proportional to the total number of genes in the gene set concerned. The names of the various GOBP terms are given in Supplementary Data 3. Bottom panels: Top 5 Gene sets from the HALLMARK collection enriched in clusters 1.1, 1.2, 2. **d** Upper panel: Boxplots of stemness indices, determined as in Malta et al.[32], in the two subtypes of retinoblastoma (subtype 1 tumors: $n = 26$, subtype 2 tumors: $n = 31$). In the boxplots, the central mark indicates the median and the bottom and top edges of the box the 25th and 75th percentiles. Whiskers are the smaller of 1.5 times the interquartile range or the length of the 25th percentiles to the smallest data point or the 75th percentiles to the largest data point. Data points outside the whiskers are outliers. Significance was tested by a two-sided Wilcoxon test, $p = 1.9 \times 10^{-7}$. Bottom panel: Heatmap of stemness indices and meta-score of the most correlated and anti-correlated HALLMARK (HM) pathways and MCP-score of the most anti-correlated immune cells. Spearman's rho and $p$-value are shown in the figure. $p < 0.0001$ (****). **e** Heatmap representing expression pattern of cone- and ganglion-associated genes in the two subtypes of retinoblastoma. Statistical significance and log2 fold-change in expression between subtype 2 and subtype 1 are also shown. Adjusted.$p \geq 0.05$ (ns), adjusted.$p < 0.05$ (*), adjusted.$p < 0.01$ (**), adjusted.$p < 0.001$ (***), adjusted.$p < 0.0001$ (****). Limma moderated two-sided $t$-tests and BH correction were used. Exact $p$-values are provided in Supplementary Data 3.

as *MKI67*. Cluster 3 presented a hypoxic gene expression program, including expression of the pro-apoptotic gene *BNIP3*. Clusters 5 and 6, accounting for 10.8% (129/1198) of all cells analyzed expressed hematopoietic markers, probably corresponding to stromal cell populations. Cluster 5 expressed monocyte/microglia markers (e.g., *CD14* and *AIF1*), whereas cluster 6 expressed T-lymphocyte markers, including markers of T-cell activation (e.g., *CD3D* and *TRAC*). A visualization of the expression of markers of each cluster is shown in Supplementary Fig. 5e, together with the expression of these markers in the normal developing retina.

To analyze the genomic heterogeneity in this tumor, we inferred copy-number variations (CNVs) in each single cell from the single-cell transcriptome data (see "Methods" section) (Fig. 5d). This analysis revealed that clusters 0–4 corresponded to tumor cells (presence of genomic alterations), whereas clusters 5 and 6 corresponded to normal cells (absence of genomic alterations). Genomic alteration patterns subdivided malignant cells into three distinct cell populations: cells with multiple genomic alterations (gains of 1q, 2q, 9p, 13q, loss of 8q), cells with 2p and 10q gains, and cells with 10q gains only. All cells from clusters 0 and 2 (*CRX*+/*EBF3*+/*GAP43*+ tumor cells), and some cluster 3 cells, corresponded to the first profile (multiple alterations). Cells from clusters 1 and 4 (*CRX*+/*ARR3*+/*GUCA1C*+ tumor cells) corresponded to the last two profiles (10q gain ± 2p gain). Lastly, some cluster 3 cells corresponded to the second profile (2p and 10q gains).

The phenotypic analysis and the inferred copy-number alterations from single-cell RNA-seq data led us to conclude that the malignant cells of the subtype 2 tumor analyzed consisted of two populations, one expressing early photoreceptor/cone markers and neuronal/ganglion cell markers (clusters 0 and 2), and the other expressing early photoreceptor/cone markers and late cone markers (clusters 1 and 4). These two cell populations existed in three states, G1/S (clusters 0 and 1), G2/M (clusters 2 and 4), and hypoxic (cluster 3). A schema summarizing the interpretation of the different clusters is shown in Fig. 5e (upper panel). The *CRX*+/*EBF3*+/*GAP43*+ tumor population (clusters 0 and 2), presenting numerous genomic alterations, appeared to be genomically homogeneous. The *CRX*+/*ARR3*+/*GUCA1C*+ tumor population (clusters 1 and 4) was less unstable and consisted of two genomically different subpopulations. A tumor progression tree constructed from the genomic alterations found in the different cell populations of this tumor is proposed in Fig. 5e (bottom panel). The co-expression of *CRX*/*EBF3*/*GAP43* (early photoreceptor/cone marker and neuronal/ganglion cell markers)

was unique to tumor cells as it was absent or very rare during normal retinal development (Supplementary Fig. 5f).

The single-cell RNA-seq analysis was performed on only one retinoblastoma. Single-cell analysis of additional tumors of both subtypes are necessary to further assess retinoblastoma heterogeneity and to investigate the relationship between retinal development and tumorigenesis using trajectory inference methods such as the ones estimating RNA velocity[59,60].

**Subtype 2 tumors are associated with a higher risk of metastasis.** We then investigated whether the retinoblastomas developing metastases belonged to a specific molecular subtype. No patients in our initial series of 102 retinoblastomas cases developed metastases. We, therefore, studied an additional series of 112 primary tumors presenting high-risk pathological features (HRPFs) at diagnosis, among which 19 tumors subsequently developed metastasis. All these patients were treated at the Garrahan Hospital (Buenos Aires, Argentina). Their clinicopathological characteristics, including HRPFs, are provided in Supplementary Data 6 and summarized in Table 2.

*TFF1* belongs to a family of small secretory molecules involved in the protection and repair of the gastrointestinal tract[61]. *TFF1* is not expressed in the normal developing retina (Supplementary Fig. 6a). It was the top upregulated gene in subtype 2 tumors compared to subtype 1 tumors (fold-change = 55, adjusted $p$-value $< 10^{-12}$, Fig. 3a, Supplementary Data 3), with expression in most subtype 2 tumors but little or no expression in subtype 1 tumors (Supplementary Fig. 6b, c). These results were confirmed based on the transcriptome of two additional tumor series[16,18] (Supplementary Figs. 6b, c and 7).

We assessed TFF1 protein expression by immunohistochemistry, in 55 of the tumors from our initial series of 102 classified retinoblastomas (18 subtype 1 and 37 subtype 2 tumors). Expression of TFF1, CRX, and ARR3 are shown for representative tumors of subtypes 1 and 2 in Fig. 6a. Subtype 1 tumors displayed little or no TFF1 expression (QS ≤ 50; QS, quick score), whereas most subtype 2 tumors displayed high levels of expression (QS > 50; Fig. 6a, b, Supplementary Data 6). We then analyzed TFF1 expression in the additional series of 112 primary tumors with HRPFs including 19 metastatic cases (Garrahan series). TFF1 expression could be evaluated in 18 of the 19 primary tumors that subsequently developed metastasis. All 18 cases were positive for TFF1 (QS > 50), in contrast to the non-metastatic cases ($p = 0.00033$) (Fig. 6b and Supplementary Data 6),

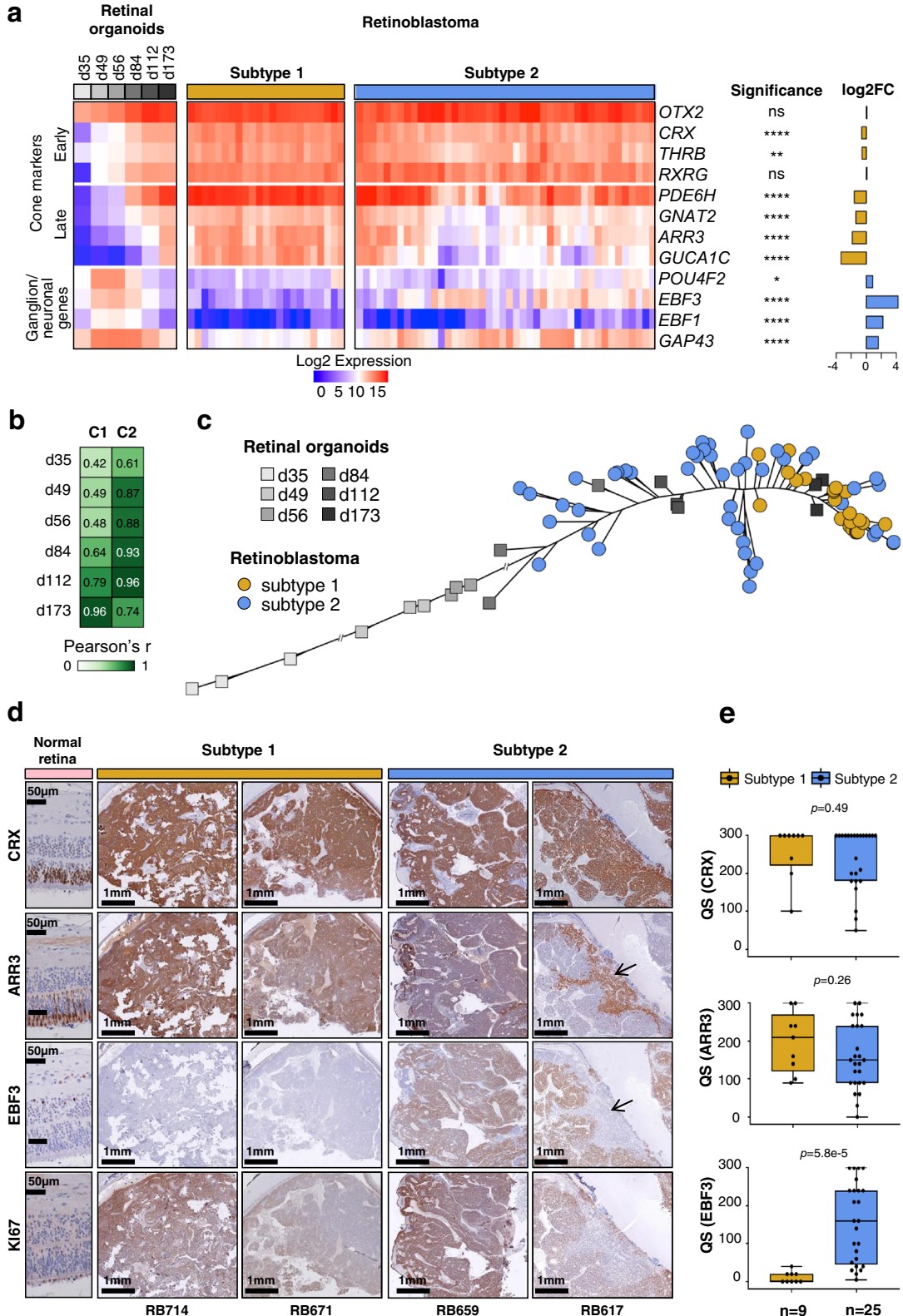

suggesting that they belonged to subtype 2. Consistent with this, 15 of the 16 metastatic cases analyzed were also positive for EBF3 (QS > 270) (Supplementary Data 6), a ganglion marker specifically associated with subtype 2 (Figs. 3a, 4d, e and Supplementary Fig. 6). In seven of the 19 metastatic cases, tissues were available from both the primary tumor and the metastasis. In all but one of these cases, the metastatic sites were also positive for TFF1 (QS range of 90–300). For EBF3, the six metastatic sites analyzed were positive (QS > 255), including the one negative for TFF1 (Fig. 6c and Supplementary Data 6). All these results suggested that subtype 2 tumors are more aggressive than subtype 1 tumors. These findings require validation by additional evidence for subtype 2 assignment, and by studies on additional series of metastatic cases.

**Fig. 4 Expression of cone and neuronal/ganglion cell markers in retinoblastoma and retinal organoids. a** Heatmap showing the expression of cone and ganglion markers in retinal organoids at different differentiation time points, and in subtype 1 and subtype 2 tumors assessed by NanoString technology. Differences in gene expression between the two subtypes were assessed by two-sided *t*-tests with BH correction. Exact *p*-values are provided in Supplementary Data 4. **b** Pearson's correlation of the expression of 8 cone markers, between the centroids of the 2 retinoblastoma subtypes and retinal organoids at different time points in differentiation. C1: centroid of subtype 1; C2: centroid of subtype 2. **c** Phylogenetic tree based on cone marker expression, for retinal organoids at different differentiation time points and for retinoblastoma samples. **d** Immunohistochemical staining of CRX, ARR3, EBF3, and Ki-67 in normal retina and retinoblastoma. For RB617, the black arrows indicate the mutually exclusive patterns for ARR3 and EBF3. Immunohistochemistry experiments were performed on 34 samples (subtype 1, *n* = 9; subtype 2, *n* = 25). Two representative images are shown for each subtype. **e** Boxplots showing the quick score (QS) for the differentiation markers used in the immunohistochemical analysis: CRX, ARR3, and EBF3. In the boxplots, the central mark indicates the median and the bottom and top edges of the box the 25th and 75th percentiles. The whiskers are the smaller of 1.5 times the interquartile range or the length of the 25th percentiles to the smallest data point or the 75th percentiles to the largest data point. Data points outside the whiskers are outliers. Two-sided Wilcoxon tests were used.

## Discussion

The use of a multi-omics approach led us to the reliable identification of two main retinoblastoma molecular subtypes. The different molecular, pathological and clinical features of these two subtypes highlighted the relevance of this classification. In support of this, we could validate the transcriptomic signatures that distinguished the two subtypes in two independent series of retinoblastoma[16,18] (Supplementary Fig. 7). The features of these two subtypes provide explanations for previous biological and clinical observations, with potential implications for retinoblastoma research and treatment.

Both subtypes expressed cone markers, consistent with the cone origin of human retinoblastoma[11–15]. There are several possible non-exclusive explanations for the existence of two subtypes of retinoblastoma. The two subtypes may be derived from cone precursors located at different retinal positions. Several studies have reported a central-to-peripheral progression of retinoblastoma location with increasing age at diagnosis[62]. As subtype 2 tumors are diagnosed significantly later than subtype 1 tumors (median age = 23.9 vs 11 months), they are therefore likely to be more peripherally located than subtype 1 tumors. The two subtypes may be derived from different cone precursors. They may also be derived from cone precursors at different stages of maturation. Arguing against this last explanation, it has been shown that $RB1^{-/-}$ retinoblastoma derived from an $ARR3^{+}$ maturing cone precursor[15].

We showed that subtype 1 tumors presented later markers of differentiated cones ($ARR3^{+}$, $GUCA1C^{+}$) and that subtype 2 tumors presented markers of earlier differentiation with an important heterogeneity between and within tumors. This is in agreement with the lower differentiation and the heterogeneity reported in older retinoblastoma patients[58]. As both subtypes are likely derived from an $ARR3^{+}$ maturing cone precursor, the lower differentiation and the heterogeneity of subtype 2 tumors with RB1 inactivation probably result from a dedifferentiation process.

We found that subtype 2 was associated with low immune and interferon response, E2F and MYC/MYCN activation, and a higher propensity for metastasis, corresponding to stemness features recently reported[32,33,35,36]. Genetic alterations and losses of function of RB1 and TP53 have also been shown to be associated with stemness in various cancers[32,36]. RB1 inactivation was present in most of the tumors of both retinoblastoma subtypes, but, nevertheless, a difference in stemness was observed between the two subtypes. The higher stemness in subtype 2 could be related to a decreased expression of another gene from the RB family, *RBL2*, located on 16q, which was lost in the majority of subtype 2 tumors. The higher stemness in subtype 2 tumors could be also related to an increased expression of *MDM4*, an inhibitor of TP53 located on 1q which was gained/amplified in 74% of subtype 2 tumors. It has been proposed that both MDM4 and

MDM2 abrogate p53-mediated tumor surveillance in retinoblastoma[63,64]. Our results indicate that MDM4 could be involved in subtype 2 tumors. In addition to the expression of cone markers, subtype 2 tumors overexpressed markers attributed to ganglion cell markers in the context of the retina. However, all these markers can also be viewed as neuronal markers (they correspond to genes expressed and involved in the central nervous system). Moreover, among the genes overexpressed in subtype 2 tumors, we identified neuronal genes expressed during the development of retinal ganglion cells but also of other retinal cell types (like *SOX11*, *DCX*, *STMN2*). These observations suggest that subtype 2 may be considered as a cone-neuronal subtype.

Expression of neuronal genes has now been found not only in the brain and neuroendocrine tumors, but also in some cancers of epithelial origin (breast, ovary, colon)[65]. In recent years, it has become clear that tumor cells exploit neuronal and neurodevelopmental pathways to proliferate, migrate, and interact with normal cells, including endothelial cells and neurons[65,66]. Therefore, the overexpression of neuronal genes that we found in subtype 2 tumors may contribute to the aggressiveness of these tumors.

The overexpression of MYCN/MYC target genes in subtype 2 tumors, and the assignment of 10 out of 11 *MYCN*-amplified tumors to subtype 2 tumors (the remaining *MYCN*-amplified tumor being unclassified) suggest that MYCN/MYC play an important role in this subtype. MYC and MYCN have been implicated in other pediatric tumors, including neuroblastoma and medulloblastoma, often in subsets of high-risk tumors. In neuroblastoma, *MYCN* amplification is found in approximately 20% of cases and is associated with high-risk disease and poor prognosis[67]. It has recently been shown that MYC could also be a driver in another subset of high-risk neuroblastomas[68,69]. Group 3 medulloblastoma are associated with *MYC* amplification (10–17%) and the worst overall survival[70,71]. The activation of MYC/MYCN in subtype 2 tumors might be exploited for specific treatments of these tumors. Indeed MYC/MYCN can be inhibited indirectly by targeting their transcription with drugs such as JQ1 and OTX015[72], or directly, by targeting MYC/MAX interaction[73].

In the series of 102 retinoblastomas, tumors with *MYCN* amplification accounted for 17% of subtype 2 tumors. *MYCN*-amplified tumors did not cluster separately from other subtype 2 tumors on transcriptome analyses, but they nevertheless had specific features. Clinically, tumors with *MYCN* amplification were diagnosed at an earlier age than other subtype 2 tumors (median age at diagnosis: 15.9 vs 26.9 months). Molecularly, the tumors with *MYCN* amplification could be distinguished from subtype 2 tumors without *MYCN* amplification on the basis of uncommon 1q gains and 16q losses. Moreover, the tumors with *MYCN* amplification were hypomethylated outside CpG islands, as in other subtype 2 tumors, but they did not display

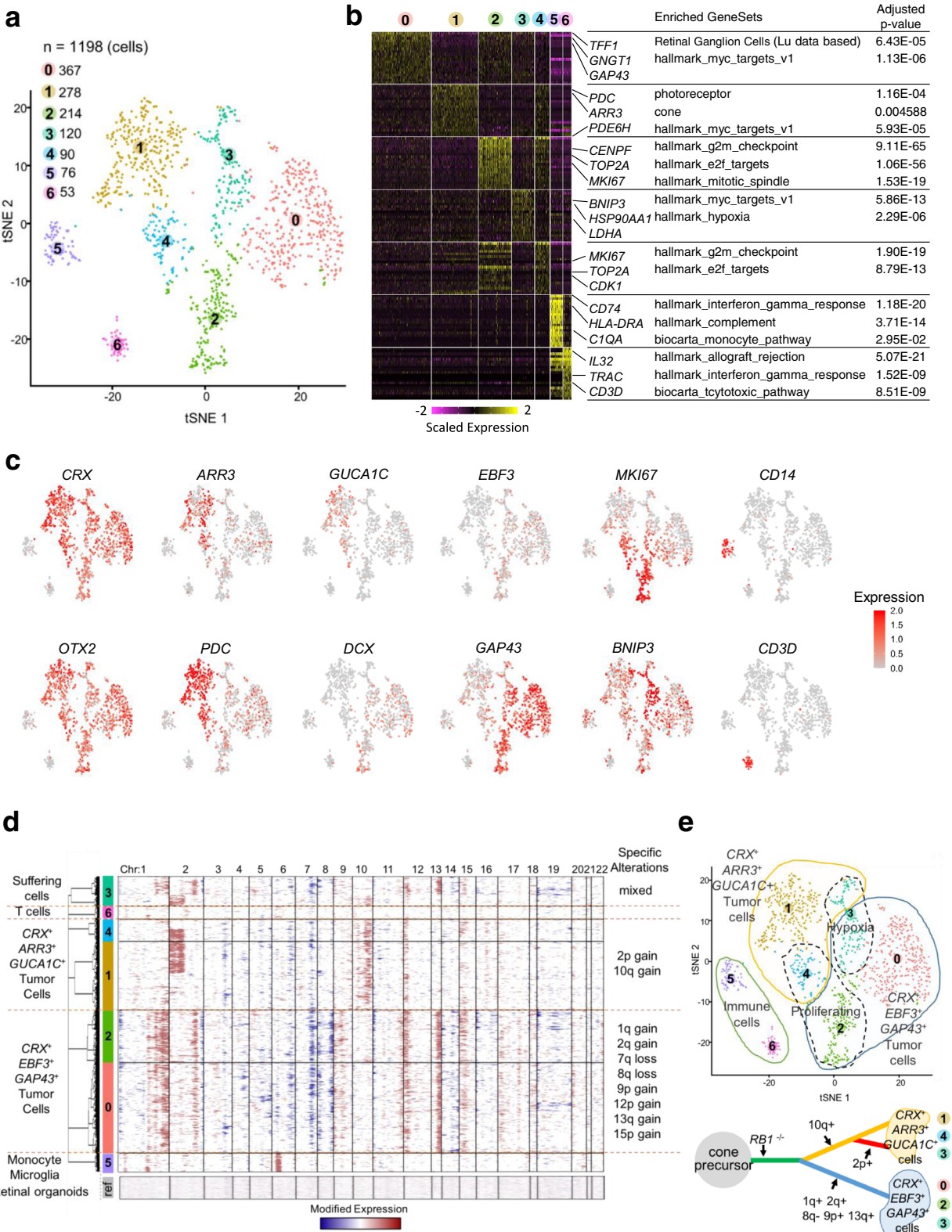

hypermethylation within CpG islands, by contrast to other sub-type 2 tumors.

In high- and middle-income countries, the frequency of enucleation for retinoblastoma is decreasing, due to early diagnosis and the development of new conservative treatments. Techniques for analyzing tumor DNA methylation and copy-number changes in aqueous humor samples and blood from cell-free DNA have

recently been developed[74,75]. The molecular characterization of retinoblastoma has, to date, been performed on tumor samples obtained from enucleation. The analyses of retinoblastoma through the use of liquid biopsy should provide a more comprehensive picture of the disease. Moreover, aqueous humor and blood samples could potentially be used to optimize retinoblastoma treatment through stratification by subtype.

**Fig. 5 Intratumor heterogeneity at the single-cell level of a subtype 2 retinoblastoma (RBSC11). a** 2D t-SNE plot of 1198 single retinoblastoma cells from one patient. Each dot represents one cell. **b** Heatmap of top cluster markers (top 20 most upregulated genes per cluster according to fold-change). Representative cluster markers and enriched gene sets are shown. Cluster marker *p*-values were calculated by hypergeometric tests with BH correction. **c** Expression of selected genes shown in 2D t-SNE plot (early photoreceptor markers: *CRX*, *OTX2*; late cone markers: *ARR3*, *GUCA1C*; neuronal/ganglion markers: *EBF3*, *GAP43*, *DCX*; proliferation marker: *MKI67*; pro-apoptotic marker: *BNIP3*; macrophage marker: *CD14*; T-cell marker: *CD3D*). **d** CNV profiles inferred from single-cell gene expression. Each row represents the profile of one individual cell. The genes on chromosome 6p overexpressed in the non-malignant cells monocyte/microglia correspond to HLA complex genes and should not be interpreted as CNV in cluster 5. **e** Upper panel: Diagram summarizing the interpretation of the different clusters of the 2D t-SNE plot. Lower panel: A progression model for this retinoblastoma case based on genomic alterations.

**Table 2 Clinical and pathological characteristics of an additional series of 112 primary tumors presenting HRPFs.**

| Characteristics | Metastatic (*n* = 19) | Non-metastatic (*n* = 93) | *p*-value |
|---|---|---|---|
| Laterality *n* (%) | | | |
| Unilateral | 14 (73.7%) | 70 (75.3%) | 0.8844[a] |
| Bilateral | 5 (26.3%) | 23 (24.7%) | |
| Age at diagnosis (months) | | | |
| Median (range) | 31 (10–88) | 31 (1–168) | 0.9166[b] |
| Initial treatment *n* (%) | | | |
| Enucleation | 15 (78.9%) | 91 (97.8%) | 0.007394[c] |
| Pre-enucleation chemotherapy | 4 (21.1%) | 2 (2.2%) | |
| IRSS Stage I HRPF | | | |
| Isolated massive choroidal invasion (+ scleral invasion) | 4 (1) (21%) | 7 (6) (7.5%) | 0.0312[c] |
| Post-laminar optic nerve invasion (+ massive choroidal and/or scleral invasion) | 9 (3) (47.4%) | 83 (49) (89.3%) | |
| IRSS Stage II | | | |
| Tumor at the resection margin of the optic nerve | 5 (26.3%) | 3 (3.2%) | 0.003428[c] |
| IRSS not classified | | | |
| Complete necrosis | 1 (5.3%) | 0 | |
| Site of metastatic relapse | | | |
| Isolated orbit | 3 (15.8%) | | |
| CNS | 6 (31.6%) | | |
| Systemic | 1 (5.3%) | N/A | |
| Orbit and lymph node | 1 (5.3%) | | |
| Orbit and systemic relapse | 3 (15.8%) | | |
| Orbit and CNS | 5 (26.3) | | |

[a]Chi² test.
[b]Two-sided Wilcoxon rank-sum test.
[c]Two-sided Fisher's exact test.

In conclusion, the identification of two molecular subtypes—cone-like and cone/neuronal—represents a major advance in the understanding of retinoblastoma. It should redefine further studies of this pediatric cancer, including the development of models, improvement of diagnosis and prognosis, and identification of more specific treatments. The high stemness and neuronal features associated with subtype 2 tumors connect retinoblastoma with emerging fields of cancer research, and open up new opportunities for treatment.

## Methods
### Patient samples
*Initial series of 102 retinoblastomas.* We included 102 tumors from 50 male patients and 52 female patients in this study. These patients came from three different hospitals: Institut Curie in Paris, France (78 patients), the Garrahan Hospital in Buenos Aires, Argentina (19 patients), and the Sant Joan de Déu Hospital in Barcelona, Spain (5 patients). The median age at diagnosis was 19.9 months (minimum: 27 days, maximum: 9.65 years). Six patients had received chemotherapy and/or radiotherapy prior to enucleation.

*Series of 112 retinoblastomas with HRPFs.* We included an independent series of 112 patients with high-risk pathological features (HRPFs)[7] from the Garrahan Hospital in this study (61 females and 51 males). The median age at diagnosis was 31 months (range: 1–168 months). Among the 112 patients, 19 subsequently developed the metastatic disease (9 females and 10 males). The median time from retinoblastoma diagnosis to metastasis was nine months (range: 4–65 months). Additional clinical characteristics are included in Table 2 and Supplementary Data 6.

Formalin-fixed paraffin-embedded tissues from the 112 tumors were analyzed. For seven metastatic patients, the metastatic sites were also available.

*Additional retinoblastoma sample for single-cell RNA sequencing.* One additional sample (RBSC11) was studied by single-cell RNA-seq. The sample was obtained from an enucleated patient >18 months of age with a unilateral non-hereditary form of retinoblastoma who did not receive treatment prior to enucleation.

*Fetal retina.* Fetal retinas were obtained from medical abortions. They were provided by the Fetal Pathology Unit of Antoine-Béclère Hospital in Paris (France). Three fetal retinas—RET215 (from a 20-week-old fetus), RET2 (23-week-old fetus), and RET1 (27-week-old fetus) were included in this study.

*Ethics statement.* All experiments were performed retrospectively and in accordance with the Declaration of Helsinki and the legislation of each participating country—France, Argentina, and Spain. The study was approved by the Institut Curie Review Board, the institutional review board of the Hospital de Pediatria Juan P Garrahan, and the Clinical Research Ethics Committee of Sant Joan de Déu Hospital. Written informed consent was obtained from parents or legal guardians of retinoblastoma patients, in accordance with current guidelines and legislation of each participating country.

Human fetuses (20, 23, 27 GW) were obtained from legally-induced terminations of pregnancy performed at the Antoine-Béclère Hospital in France. Fetal tissues were collected with the women's written consent, in accordance with the legal procedure agreed by the French National Agency for Biomedical Research (Agence de Biomédecine) and the approval of the local ethics committee of Antoine-Béclère Hospital.

**Human iPSC maintenance and retinal organoid generation.** Human-induced pluripotent stem cells (iPSCs) derived from dermal fibroblasts (hiPSC-2 clone)[52]

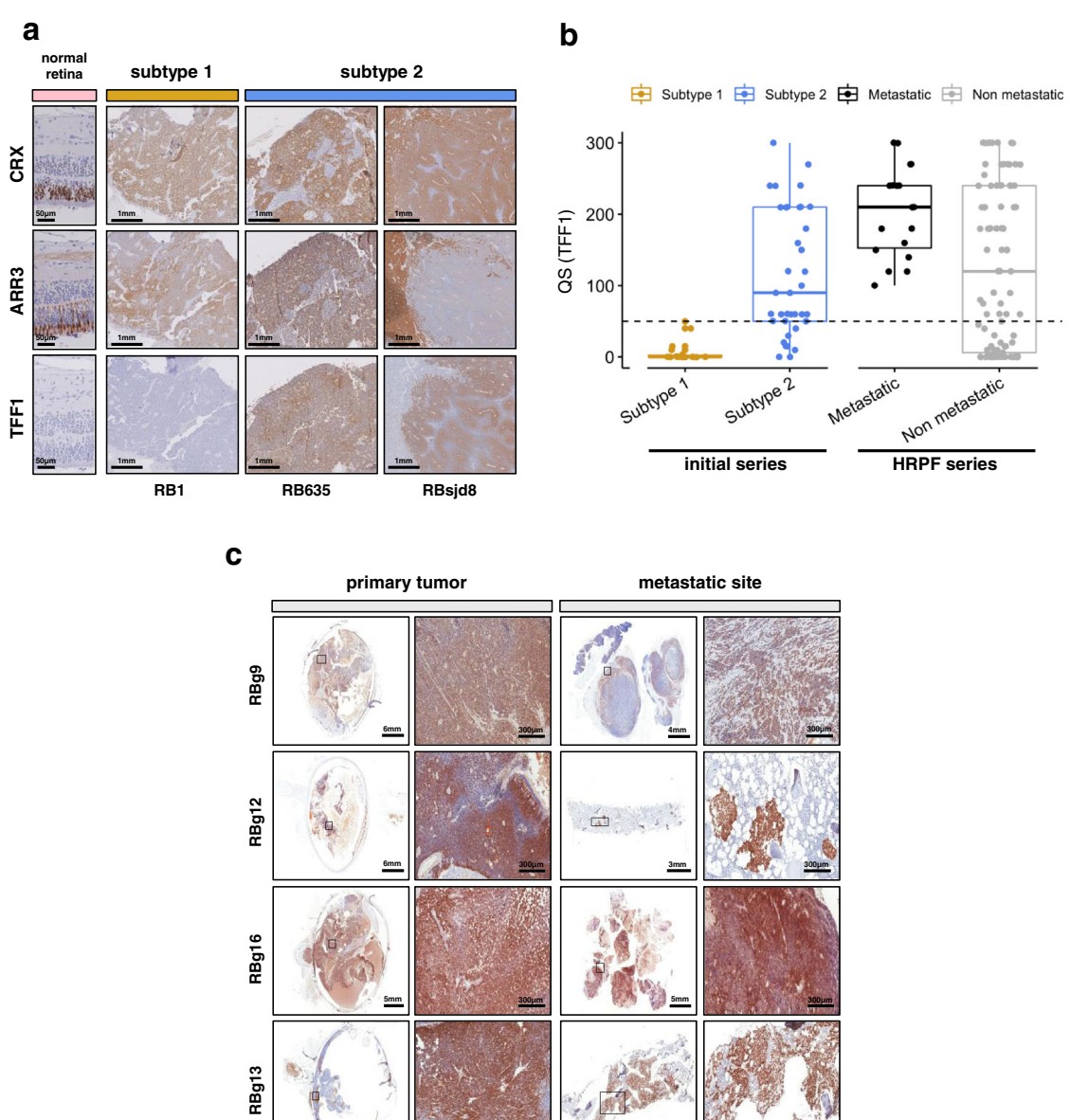

**Fig. 6 Subtype 2 tumors are associated with a higher risk of metastasis. a** Immunostaining of CRX, ARR3, and TFF1 in normal retina and retinoblastoma. Immunohistochemistry experiments were performed on 55 samples (subtype 1, $n = 18$; subtype 2, $n = 37$) from the initial series of 102 retinoblastomas. Representative images are shown: one subtype 1 tumor (RB1) and two subtype 2 tumors (RB635, RBsjd8). The subtype 2 tumors presented either a co-staining (RB635) or a mirror pattern (RBsjd8) for ARR3 and TFF1. **b** Boxplots showing the quick score (QS) for TFF1 in 55 tumors of the initial series (subtype 1, $n = 18$; subtype 2, $n = 37$), and in 112 tumors of the HRPF series. In the boxplots, the central mark indicates the median and the bottom and top edges of the box the 25th and 75th percentiles. The whiskers are the smaller of 1.5 times the interquartile range or the length of the 25th percentiles to the smallest data point or the 75th percentiles to the largest data point. Data points outside the whiskers are outliers. Two-sided Wilcoxon tests were used to assess the difference of the QS for Subtype 1 vs Subtype 2, $p = 1.1 \times 10^{-7}$, and metastatic vs non-metastatic, $p = 0.007$. **c** Immunostaining of TFF1 for primary tumors of metastatic retinoblastoma (left) and their metastatic sites (right), at low and high magnification. TFF1 expression could be assessed by immunohistochemistry for 6 of 7 available primary-metastasis tumor pairs. Representative images of four are shown.

were cultured on truncated recombinant human vitronectin-coated dishes in a humidified 37 °C incubator with 5% $CO_2$ in Essential 8TM medium (ThermoFisher Scientific) with daily medium change and weekly passage (2 ml enzyme-free Gentle cell dissociation reagent for 7 min at room temperature)[48]. For retinal differentiation, adherent iPSCs were expanded to 70–80% and cultured in Essential 6^TM medium (ThermoFisher Scientific) for 2 days, followed by replacing each 2–3 days Essential 6^TM medium supplemented with 1% N-2 Supplement, 10 units/ml Penicillin and 10 μg/ml Streptomycin (ThermoFisher Scientific). At around day 28, retinal organoids were isolated with a needle and cultured as floating structures in ProB27 medium (DMEM:Nutrient Mixture F-12 1:1, L-glutamine, 1% MEM non-essential amino acids, 2% B27 supplement (ThermoFisher Scientific), 10 units/ml Penicillin, and 10 μg/ml Streptomycin) supplemented with recombinant human FGF2 (PeproTech) for a week and then in ProB27 medium for the next several weeks allowing retinal differentiation and maturation[48,76]. By RT-qPCR and

immunofluorescence analysis, we previously showed that the different iPSC lines (hiPSC-2 clone[52], AAVS1:CrxP_H2BmCherry hiPSC line[77]) we used, are able to produce the whole repertoire of retinal cells, in an identical way and following the same chronological order with first the appearance of ganglion cells, then the amacrine and horizontal cells and finally the mature photoreceptors, the bipolar cells, and the Müller glial cells. The use of different markers of photoreceptor lineage (CRX, RCVRN, NRL, NR2E3, ARR3, RHO, OPSINs…) showed that the genesis of cones and rods is identical in the different iPSC lines used.

### Sample collection and processing
*Tumor samples. Institut Curie.* Immediately after enucleation, a needle was inserted through the anterior chamber of the eye to extract a tumor sample by aspiration. The tumor specimen was placed in an RPMI medium on ice. The cells were

resuspended, counted and the suspension was split in two (for DNA and RNA preparation). The tubes were then centrifuged to remove the medium and the pellet was snap-frozen for later extraction. The remainder of the ocular globe was paraffin-embedded. For tumor DNA extraction, samples were first incubated in lysis buffer with recombinant proteinase K (Roche, Boulogne-Billancourt, France). They were next incubated with RNAse A (Roche). DNA was then extracted using a standard phenol–chloroform protocol. Tumor RNA was extracted using the miRNeasy Mini Kit, according to the manufacturer's instructions (Qiagen, Courtaboeuf, France).

*Garrahan Hospital and Sant Joan de Déu Hospital.* Immediately after enucleation, a needle was inserted through the anterior chamber of the eye to extract a tumor sample by aspiration. The tumor specimen was either placed in guanidine thiocyanate or snap-frozen for later extraction. For tumor samples preserved in guanidine thiocyanate, alkaline phenol/chloroform/isoamyl alcohol (24:1:25) extraction was used for tumor DNA extraction. For snap-frozen tumor samples, commercial affinity columns (QIAamp DNA Mini Kit, Qiagen) or a standard phenol–chloroform protocol were used for tumor DNA extraction.

*Single-cell RNA-seq sample.* Tumor sample was processed immediately following needle aspiration through the anterior chamber of the eye. The sample was placed in an ice-cold $CO_2$-independent medium. Density gradient centrifugation by Histopaque-1077 (Sigma-Aldrich) was used to remove debris, dead cells, and erythrocytes. The isolated viable cells were mechanically dissociated, washed, and resuspended in phosphate-buffered saline supplemented with 0.04% bovine serum. Cell count and viability were determined by trypan blue exclusion on a Vi-CELL XR (Beckman Coulter Life Sciences).

*Blood samples.* For Curie hospital samples, normal DNA was extracted with a perchlorate/chloroform protocol or FujiFilm QuickGene technology (Kurabo Biomedical, Osaka, Japan). For Garrahan Hospital samples, normal DNA was extracted with a phenol/chloroform/isoamyl alcohol (24:1:25) protocol or with commercial affinity columns (QIAamp DNA Mini Kit, Qiagen). For Sant Joan de Déu Hospital samples, a standard isopropanol precipitation protocol was used.

*Fetal retina.* Fetal tissues were maintained in ice-cold Hanks balanced salt solution (HBSS) after medical abortions. For the isolation of neural retinal tissue, eyes were transferred onto a sterile Petri dish containing ice-cold PBS and maintaining a cornea side-up position with fine forceps. A small incision was made in the corneoscleral junction using a small scalpel. The tip of the curved microscissors was inserted into the small incision. Eyes were carefully rotated of 360 degrees, and small incisions were made all the way around the eye, parallel to the corneoscleral junction, allowing dissociation of the anterior eyecup and lens from the posterior eyecup. The posterior eyecup was passed onto a small Petri dish containing ice-cold PBS. The neural retina was carefully isolated from the underlying retinal pigmented epithelium by blunt dissection using fine forceps. RNA was extracted using the miRNeasy Mini Kit, according to the manufacturer's instructions (Qiagen, Courtaboeuf, France).

*Human iPSCs.* Total RNA was extracted from human iPSCs using the Nucleospin RNA II kit (Macherey-Nagel), according to the manufacturer's instructions.

**Gene expression arrays.** RNA of 59 samples (see Supplementary Data 1) were hybridized, in two batches, to Affymetrix Human Genome U133 plus 2.0 Array Plates (Santa Clara, CA) according to Affymetrix standard protocols. Raw CEL files were RMA[78] normalized using R package affy 1.60.0. Batch effects were corrected with the help of the Bioconductor package SVA 3.30.1. The arrays were mapped to genes with a Brainarray Custom CDF (EntrezG version 23)[79]. Independent component analysis in $k = 3$ independent components (IC) was performed using R package MineICA 1.24.0 (JADE method)[80,81]. The genes with high negative ($< −2.5$) or positive contributions ($>2.5$) to IC were analyzed through pathway enrichment analysis (hypergeometric tests), seeking specifically signatures related to potential contamination by stromal cells. Genes with high positive contributions to IC #1 were found highly enriched in markers of stromal cells, and were discarded from clustering analyses.

**DNA methylation arrays.** Sixty-six DNA samples (Supplementary Data 1) were hybridized on Infinium HumanMethylation450 BeadChip arrays (Illumina, San Diego, CA). Four microliters of bisulfite-converted DNA were used for hybridization, following the Illumina Infinium HD Methylation protocol[82]. Data were processed using preprocessIllumina and getBeta functions in R package Minfi 1.28.4[83]. Probes were annotated using the R package IlluminaHumanMethylation450kmanifest 0.4. Probes located in Chromosome X and Chromosome Y were discarded from subsequent analyses.

**SNP arrays and BAC-CGH arrays.** Ninety-five retinoblastomas were analyzed using SNP arrays or BAC-CGH arrays (Supplementary Data 1). Seventy tumor samples were analyzed on high-density SNP arrays. The B allele frequency and log-ratio signals were smoothed and analyzed using the Genome Alteration Print (GAP) algorithm (http://bioinfo-out.curie.fr/projects/snp_gap/)[84]. Twenty-five

tumor samples were analyzed on BAC-CGH microarrays. These arrays consisted of 3510 or 5323 clones covering the human genome with an average resolution of 850Kb or 560Kb; they were designed by the CIT-CGH Consortium (INSERM U830, Institut Curie, Paris) and IntegragenTM[85]. Hybridized slides were scanned and the scan data was pre-processed using R package MANOR 1.36.0[86] to correct for local spatial bias and continuous spatial gradient. Each array-CGH profile was centered on the median log2 ratio and then analyzed to extrapolate copy-number profiles using the GLAD algorithm 2.28.1[87].

**Whole-exome sequencing.** Whole-exome sequencing was performed for 71 retinoblastomas and matched normal (blood) samples (Supplementary Data 2). For 32 tumor/normal sample pairs, sequence capture and exome sequencing were performed by the Sequencing Platform of Institut Curie. The Nextera exome enrichment kit (Illumina) was used for DNA library preparation. The eluted fraction was amplified by PCR and sequenced on an Illumina HiSeq 2500 sequencer as paired-end $100 \times 100$ bp or $150 \times 150$ bp reads. For the remaining 39 tumors/normal sample pairs, sequence capture and exome sequencing were performed by Integragen. The protocol followed by Integragen has been described elsewhere[88]. In brief, Agilent in-solution enrichment (SureSelect Human All Exon Kit v4 + UTR) was used for DNA library preparation. The eluted fraction was amplified by PCR and sequenced on an Illumina HiSeq 2000 sequencer as paired-end 75 bp reads.

**Single-cell library preparation and sequencing.** Six thousand cells were loaded onto the Chromium System using the single-cell 3′ reagent kits v2, in accordance with the manufacturer's protocol (10× Genomics), where single cells are partitioned in droplets. Following capture and lysis, cDNA incorporating UMI (unique molecular identifier) and cell barcode was synthesized and amplified. Amplified cDNA was fragmented and the Illumina sequencing library was constructed as per the manufacturer's protocol (Illumina). Libraries were loaded at 400pM and paired-end sequenced on Novaseq 6000 using NovaSeq 6000 S1 Reagent Kit (Illumina). Cells were sequenced at a mean depth of 100000. For quality control and quantification of cDNA and library, BioAnalyzer (Agilent BioAnalyzer High Sensitivity chip) was used.

**Additional RNA quantification, DNA methylation, and mutation analyses**

*NanoString® codeset design and mRNA quantification.* A codeset of 22 target genes was custom-designed and manufactured by NanoString® (Supplementary Data 4). One hundred nanograms of total RNA extracted from each sample was assessed on the Gen2 nCounter Analysis System from NanoString® Technologies at the Genomics Platform of the Curie Institute following the manufacturer's instructions. Samples were hybridized with multiplexed NanoString® probes containing a biotinylated capture probe and a reporter probe attached to a fluorescent barcode specific for each transcript, according to the nCounter codeset design (NanoString, Seattle, WA, USA). Hybridized samples were then purified and immobilized in a sample cartridge on the nCounter Prep Station for data collection, followed by quantification of the target mRNA in each sample using the nCounter Digital Analyzer (NanoString®). Data were normalized according to NanoString guidelines with nSolver 4.0. Briefly, the background was subtracted using the geometric mean of negative controls provided by NanoString®. The matrix was log-transformed (base 2) for further analysis.

*Pyrosequencing.* Forty-seven retinoblastoma samples were analyzed by performing pyrosequencing of the 9 selected CpGs (Supplementary Data 1 and Data Analysis section (Array-based methylation signature)).

Bisulfite treatment of genomic DNA (500 ng) was performed using the EZ DNA Methylation kit (Zymo Research). Primer design for each CpG target was performed using the PyroMark Assay Design software 2.0.2 (Qiagen) and pyrosequencing reaction was performed using PyroMark Q24 instrument (Qiagen). Primers used are provided in Supplementary Data 7. Pyrograms obtained were analyzed using the PyroMark Q24 software 2.0.6.20 (Qiagen) and methylation status was calculated at each CpG of interest.

*Targeted sequencing.* Targeted sequencing of the exonic regions of *RB1*, *BCOR,* and *ARID1A* was performed by IntegraGen SA (Evry, France) on 23 samples from the series of 102 retinoblastomas not subjected to whole-exome sequencing (Supplementary Data 2). The Fluidigm Access Array microfluidic system was used. PCR products were barcoded, pooled, and subjected to Illumina sequencing on a MiSeq instrument as paired-end 150-bp reads.

*Sanger sequencing.* Primer design was performed using Primer3 plus software[89]. Their sequences are provided in Supplementary Data 7. PCR amplification was performed with the HotStarTaq plus DNA Polymerase (Qiagen). PCR products were purified and sequenced at the Genomics Platform of the Institut Curie, using an ABI 3730 XL (Applied Biosystems, Life Technologies). Sequence analysis was carried out using Sequencher® version 5.4.1 sequence analysis software (Gene Codes Corporation, Ann Arbor, MI USA). One hundred nonsynonymous variants were identified by whole-exome sequencing and all variants identified by targeted sequencing were verified using Sanger dye-terminator sequencing. We validated 92

nonsynonymous mutations identified by whole-exome sequencing (of 100 variants tested, 92%) and all the mutations identified by targeted sequencing.

**Immunohistochemistry**. Immunohistochemical staining was performed on 3 μm-thick sections.

For the cohort of 102 retinoblastomas included in this study, automated immunostaining for CRX, ARR3, EBF3, Ki-67 (Supplementary Data 4), and TFF1 (Supplementary Data 6) was performed on the available paraffin-embedded samples with Autostainer 480 (Lab Vision) at Institut Curie. The following antibodies were used: anti-CRX (Abcam, ab140603; 1:300 for AFA/Bouin fixed tissue and 1:500 for formalin-fixed tissues), anti-ARR3 (Proteintech Group, 11100-2-AP; 1/300 for AFA/Bouin fixed tissue and 1/500 for formalin-fixed tissues), anti-EBF3 (Abnova Corporation, H00253738-M05; 1/800), anti-Ki-67 (Abcam, ab1558; 1/2500), and anti-TFF1 (Sigma-Aldrich, HPA003425; 1/1000). Additional information about the conditions used is described in Supplementary Data 4. For each slide, staining was assessed by eyeballing independently by two specialists (authors: NS and PF) blind to molecular subtype classification, taking into account the intensity (I) as null (0), mild (1), moderate (2), and strong (3), and the percentage (P) of tumor cells with stained nuclei for CRX and EBF3 and stained cytoplasm for ARR3 and TFF1. The quick score (QS) was then calculated as I * P (from 0 to 300).

For the independent series of 112 retinoblastomas with high-risk pathological features from Garrahan Hospital, immunostaining of TFF1 was performed manually in the Pathology Department of the Garrahan Hospital according to the procedure used at Institut Curie. For each slide, staining was assessed independently by three specialists (authors: R.A., F.L., and G.L.).

## Bioinformatics and data analysis

*GISTIC analysis.* The copy-number alteration data for the 72 retinoblastomas studied by consensus clustering were first analyzed using GISTIC2.0 2.0.22[90]. Twelve significant recurrent copy-number alteration regions were identified. The average copy number for each sample across these regions was then used for consensus clustering of the copy-number alteration data.

*Consensus clustering.* Consensus clustering was performed independently on the transcriptomic, methylomic, and GISTIC-processed copy-number alteration data of 72 retinoblastoma samples ($n = 59$ transcriptomes, $n = 66$ methylomes, $n = 72$ copy-number alteration profiles) (Supplementary Data 1). mRNA expression was assessed through Affymetrix U133plus2.0 arrays, genome methylation through Illumina Infinium Human Methylation 450 BeadChip arrays, and somatic copy-number alterations through SNP arrays or CGH-BAC arrays.

For the transcriptomic data, consensus hierarchical clustering was derived from a series of 24 dendrograms, which were obtained on all 59 retinoblastoma samples (columns) by analyzing 8 data subsets related to various numbers of genes (rows), through hierarchical clustering using 3 different linkage methods (average, complete, and Ward) and one distance metric (1 − Pearson correlation coefficient). To construct the 8 data subsets, various number of genes (rows) (spanning between 100 and 4709 genes) were selected based on 2 criteria: minimal robust coefficient of variation (rCV) thresholds spanning the 99.5th to the 60th percentiles, and p-value lower than 0.01 for a test of variance (we test whether the variance for a gene is higher or not than the median variance across all genes).

Having obtained these 24 dendrograms, we cut each dendrogram in $k$ clusters, and get a series of partitions in $k$ groups, for $k$ ranging from 2 to 8 (NB: a partition in $k$ groups is called a $k$-partition). For each value of $k$, we then derived a consensus $k$-partition from the 24 $k$-partitions obtained from the 24 dendrograms. To do so, we first calculated the (samples × samples) co-classification matrix from these 24 $k$-partitions (NB: in the co-classification matrix, the cell $(i,j)$ reports the number of partitions where samples $i$ and $j$ belong to the same group). The co-classification matrix is a similarity matrix and can be transformed into a dissimilarity matrix by replacing the value x in each cell $(i,j)$ by MAX_VALUE – $x$ (Here MAX_VALUE = 24). Then this dissimilarity matrix can be used to perform the hierarchical clustering of the related samples, using the complete linkage. Finally, the obtained dendrogram is cut in $k$ clusters to yield the consensus k-partition.

Of note, before calculating the consensus $k$-partitions ($k$ from 2 to 8), we assessed the intrinsic stability of the underlying k-partitions, as compared to k-partitions obtained using the same linkage and the same set of genes, but based on "noisy" data. "Noisy" data were generated for each of the 8 data subsets (200 iterations for each) by addition of random Gaussian noise ($\mu = 0$, $\sigma = 1.5\times \times$ median variance calculated from the data set). The stability of each initial $k$-partition was then assessed using a stability score corresponding to the mean symmetric difference distance between an initial $k$-partition and the corresponding $k$-partitions derived from "noisy" data. The symmetric difference distance compares two partitions and gives the proportion of retention of the pairs of samples that are in the same group. It brings values ranging from 0 to 1: comparing two equal partitions yields a value of 1.

Consensus clustering of the methylomic data ($n = 66$ retinoblastomas) was performed in a similar manner, this time with between 2086 and 87937 CpGs selected (rCV thresholds spanning the 99.5th to the 60th percentiles and a p-value lower than 0.01 for the test of variance). Consensus clustering of the GISTIC-processed copy-number alteration data ($n = 72$ retinoblastomas) was also performed in a similar manner, this time with 3 or 4 significant copy-number

regions selected (rCV thresholds spanning the 80th to the 50th percentiles and a p-value lower than 0.01 for the test of variance). We observed both for transcriptome and methylome that the (intra-omics) consensus partition with $k = 2$ clusters was more stable than solutions with $k > 2$ clusters. We thus selected $k = 2$ clusters for these two omics. The DNA copy-number data yielded 5 clusters.

*Cluster-of-clusters and centroid classification.* To identify a common samples' partition across all three genomic platforms (transcriptome, methylome, copy number), we used a cluster-of-cluster approach. Based on the three unsupervised consensus partitions previously obtained from the three omics datasets (one consensus partition per omics data set), we first built a (samples × samples) co-classification matrix, with values ranging from 0 to 1, with 0 corresponding to a pair of samples that never co-classify in any genomic data set, and 1 corresponding to a pair of samples that always co-classify in all three genomic datasets. This matrix was then subjected to hierarchical clustering using complete linkage. Three clusters of clusters were thus identified ($n = 27$, $n = 37$, and $n = 8$). The two larger cluster-of-clusters corresponded to two core molecular subtypes, subtype 1 and subtype 2. The smallest cluster-of-clusters ($n = 8$) corresponded to ambiguous samples whose cluster assignments were not consistent across all three genomic platforms.

To classify these remaining samples according to either subtype 1 or subtype 2, we built two supervised centroid-based predictors, one transcriptomic and the other methylomic. The two core clusters of clusters defining subtype 1 and 2 were used to train these classifiers. For the transcriptomic data, the centroids of subtype 1 and subtype 2 were calculated as the intra-cluster median expression of the 800 genes most significantly differentially expressed between the two clusters (taking the 400 most upregulated genes in each subtype); similarly, for the methylomic data, the centroids of subtypes 1 and 2 were based on the median beta value of the 10,000 CpGs most significantly differentially methylated between the two clusters (5000 most methylated in each subtype). Each sample was assigned to the class whose centroid was closest to its profile, based on a Pearson's correlation coefficient of at least 0.1 (we let unclassified samples yielding a Pearson's correlation coefficient less than 0.1 to any of the two centroids/classes). Following this centroid-based step, we could classify 6 of the 8 samples without initial cluster-of-cluster attribution (four were assigned to cluster 1, two to cluster 2). This step also identified 3 outlier samples: two were already unclassified after the first cluster-of-clusters step, one was attributed initially to cluster 2.

*Copy-number analysis.* Copy-number alterations (CNAs) were analyzed using whole-exome sequencing (WES) data ($n = 63$), SNP arrays (Illumina HumanCNV370 quad, $n = 15$; Illumina Human610 quad, $n = 6$; Affymetrix Cytoscan, $n = 3$), and BAC arrays (3510 markers, $n = 12$; 5323 markers, $n = 3$). BAC arrays were analyzed using GLAD algorithm 2.28.1[87] to smooth log-ratio profiles into homogeneous segments and assign a discrete status to each segment (homozygous deletion, deletion, normal, gain, amplification). SNP arrays were analyzed using the Genome Alteration Print method[84], which takes into account both the log ratio and B allele frequency signals to determine normal cell contamination, tumor ploidy, and the absolute copy-number of each segment. The median absolute copy-number was considered to be the zero level of each sample. Segments with an absolute copy number > zero + 0.5 or < zero − 0.5 were considered to have gains and deletions, respectively. Segments with an absolute copy-number ≥5 or ≤0.5 were considered to have high-level amplifications and homozygous deletions, respectively. To identify CNAs using WES data, we calculated the log ratio of the coverage in each tumor and its matched normal sample for each bait of the exome capture kit with a coverage ≥ 30× in the normal sample. Log-ratio profiles were then smoothed using the circular binary segmentation algorithm, as implemented in the Bioconductor package DNAcopy 1.50.1[91] (default parameters except min.width = 4, undo.splits = sdundo, undo.SD = 1.5). The most frequent smoothed value was considered to be the zero level of each sample. Segments with a smoothed log ratio >zero + 0.15 or <zero − 0.15 were considered to have gains and deletions, respectively. High-level amplification and homozygous deletion thresholds were defined as the mean ± 5 s.d. of log ratios in regions of normal copy number. Visual inspection of the profiles allowed to validate recurrent focal amplifications and homozygous deletions.

For a given sample, the GNL (Gain = 1/Normal = 0/Loss = −1) copy-number data are aggregated by chromosome, as the proportion of features with an aberration (i.e., gain or loss). The overall genomic instability score corresponds to the mean score across all chromosomes.

**Whole-exome sequencing analysis pipeline and mutation annotation.** Sample reads were aligned using Burrows–Wheeler Aligner (BWA 0.7.4)[92]. Targeted regions were sequenced to an average depth of 82×, with 99% of the regions covered by ≥1×, 97.0% covered by ≥4×, and 87% covered by ≥20×.

For detection of somatic single-nucleotide variants (SNVs) and base insertions or deletions (indels), we used two separate variant-calling pipelines, the results of which were then merged. The first pipeline used MuTect 1.1.5[93] for SNV calling and the GATK SomaticIndelDetector 2.1–8 for indel calling[94–96]. The second pipeline used VarScan 2.3.7 somatic and VarScan somatic filter for both SNV and indel calling (http://varscan.sourceforge.net)[97]. After the variants called by both pipelines were merged, they were annotated using Annovar v2014Mar10[98]. Custom

filters and manual curation using the Integrative Genomics Viewer (IGV 2.3.34)[99] were then used to maximize the number of true positive calls and to minimize the number of false positives.

### Methylation analysis

*Array-based methylation signature.* From the methylome array data ($n = 66$), we selected the most differentially methylated CpGs between the two retinoblastoma subtypes (clusters of clusters) based on statistics of the Wilcoxon test. Out of the top 50 hypermethylated CpGs and top 30 hypomethylated CpGs of subtype 2 retinoblastoma (by $p$-value), top 7 hypermethylated and top 7 hypomethylated CpGs by the difference of beta value were selected for pyrosequencing. 5 of them did not perform well in pyrosequencing. This method led to the selection of 9 CpGs significantly differentially methylated that have been analyzed by pyrosequencing for sample classification. Seventeen samples from the initial series were analyzed by pyrosequencing for validation of the nine-CpG-based classifier (9 subtype 1, 8 subtype 2); from these samples, we derived subtype 1 and subtype 2 centroids based on these 9 CpGs. The nearest-centroid approach (with Pearson's metric and a minimal threshold of 0.3) correctly assigned 16 of these 17 samples to their known subtype and left unassigned the remaining sample. Additional samples analyzed by pyrosequencing for these 9 CpGs were then classified using the nearest-centroid approach (Pearson's metric) at a minimal threshold of 0.3.

*Differential methylation analysis.* Differential methylation analysis was performed by two-sided Wilcoxon rank-sum test and BH correction to compare methylation level of 473,864 probes between 27 subtype 1 and 36 subtype 2 retinoblastomas. 94,101 probes were found differentially methylated between the two subtypes (69,901 probes higher in subtype 1, 24,200 probes higher in subtype 2). 6607 probes had a difference of beta value of more than 0.2 (4520 higher in subtype 1, 2087 higher in subtype 2) (Supplementary Data 2).

### Differential gene expression and pathway enrichment analysis.

Differential gene expression analysis was performed by Limma R package 3.40.6[100] to compare the expression of 20,408 genes between 26 subtype 1 and 31 subtype 2 tumors. 6207 genes were found differentially expressed (adjusted $p$-value < 0.05) (Supplementary Data 3). Three main gene clusters were identified by hierarchical cluster analysis (mean centering of genes, $1 -$ Pearson's correlation coefficient as distance and average linkage). Visualization using heatmaps was performed with the R package ComplexHeatmap 2.1.1. Pathway enrichment analysis was performed by R clusterProfiler package 3.12.0[101]. Enriched gene sets from GOBP (Gene Ontology Biological Process) with adjusted $p$-value < 0.01 were selected for CytoScape (3.7) EnrichmentMap (2.1.1) analysis[102]. Gene sets tested (GOBP and HALLMARK) were from the Molecular Signatures Database (MSigDB, version 6.2)[103].

### Evaluation of stemness by transcriptome.

Stemness indices in retinoblastoma were evaluated as described in Malta et al.[32]. Briefly, the weight vectors of 12,955 genes were obtained by Malta et al. as a stemness signature to identify pluripotent stem cells from progenitor cells in PCBC (Progenitor Cell Biology Consortium) transcriptomic data set. 12,364 genes were available in our data set. After mean-centering, the expression matrix, Spearman's correlation with the stemness signature vectors was calculated for each sample of retinoblastoma and then scaled to the range of 0 to 1 as the stemness indices. The other three stemness indices were estimated using three stemness gene signatures (Miranda et al., Shats et al., Smith et al. of 109, 80, and 49 genes, respectively)[33,35,36] by ssgsea function of R package gsva 1.30.0. Boxplots were generated using R package ggpubr 0.2.0.

### Pathway meta-score.

Pathway meta-scores were calculated as the average expression of the genes involved in one selected pathway and then centered and scaled.

### Analysis of two independent transcriptomic datasets.

We applied the nearest-centroid approach (with Pearson's metric and a minimal threshold of 0.1) using the transcriptomic centroids calculated from our datasets to classify two publicly available transcriptomic datasets (GSE59983 and GSE29683).

In the Kooi et al.'s series[18] ($n = 76$), 46 subtype 1 samples and 28 subtype 2 samples were identified, 2 samples were unable to be assigned a subtype. In the McEvoy et al.'s series[16] ($n = 55$), 24 subtype 1 samples and 22 subtype 2 samples were identified among the 48 samples, 2 samples were unable to be assigned a subtype. Some samples ($n = 7$) were excluded from clustering analysis due to the high contamination of retinal pigmental epithelial (RPE) cells. We examined the average expression of an RPE gene signature (from Liao et al.[104], $n = 83/87$ genes present in the data) and removed the suspected outlier samples ($n = 7$) by Interquartile rule (suspected outliers are the samples when their average expression of RPE signature > $Q3 + 1.5$ IQR or < $Q1 - 1.5$ IQR).

### Phylogenetic analysis of retinoblastoma and retinal organoids.

Gene expression data of 8 genes related to cone-cell differentiation (*OTX2, CRX, THRB, RXRG, PDE6H, GNAT2, ARR3, GUCA1C*) were assessed by NanoString in 67 retinoblastomas (23 subtype 1 and 44 subtype 2) and 18 retinal organoids at 6 time

points after induction from iPSCs were used in phylogenetic analysis. Phylogenies were inferred by the minimal evolution algorithm[105] using fastme.bal function in R ape package 5.3 applied to Euclidean distance matrix based on these 8 gene expressions.

### Single-cell transcriptome analysis

*RBSC11 retinoblastoma.* Sample demultiplexing, alignment to the reference genome (GRCh38, Ensembl 84, pre-built Cell Ranger reference version 1.2.0), quantification and initial quality control (QC) were performed using the Cell Ranger software (version 2.1.1, 10× Genomics).

Genes that were expressed in more than 3 cells and cells that expressed more than 500 genes and less than 5% of mitochondria genes were retained ($n = 1198$). The median numbers of genes and UMI counts per cell were 2911 and 7749, respectively. Normalization and clustering were performed using Seurat package version 2.3.4. UMI counts were normalized by NormalizeData function with logNormalize method, by a scaling factor of the median UMI count. UMI counts were then scaled to regress out the effect of UMI counts. Variable genes were found with FindVariableGenes function with logVMR function. Genes with an average expression more than 0.0125 and <8 and with dispersion more than 0.5 were considered as variable genes for principal component analysis (PCA). Cell clusters were identified by FindClusters function with shared nearest neighbor (SNN) method modularity optimization-based clustering algorithm[106], using the first 20 principal components. The parameter Resolution in the FindClusters was set between 0.4 and 1.4 and finally set to 0.6 for it provided a better biological interpretation.

Cluster markers were identified by FindAllMarkers function. Briefly, the expression of genes that expressed in more than 10% of cells in one cluster were compared with the expression of these genes in all other clusters, using Wilcoxon rank-sum test and corrected with BH correction. The procedure was repeated for all clusters. Genes upregulated in each cluster with more than 0.2 fold were considered as cluster markers. Pathway analysis of cluster markers was performed by R clusterProfiler package[101]. Gene sets tested were from the Molecular Signatures Database[103] (HALLMARK and BioCarta) and from Supplementary Data 3 (Cell type markers_Lu data and Selected cell type markers).

Correlation to bulk mRNA expression profiles of purified cell types was performed by R SingleR package 1.0.1[107]. The expression profile of each cell was compared with the expression profiles of a data set that contains 713 microarray samples classified to 38 main cell types and further annotated to 169 subtypes[108].

Copy-number variations (CNVs) were inferred from the single-cell gene expression by InferCNV package 0.8.2, using normal retinal organoids derived from hiPSCs as reference.

*Normal developing retina (Lu et al.[38], data).* Normal retina scRNA-seq data from Lu et al.[38] were retrieved from GEO Omnibus database GSE138002. We retrieved the final filtered count matrix (GSE138002_Final_matrix.mtx.gz), gene annotations (GSE138002_genes.csv.gz), and cell annotations (GSE138002_Final_barcodes.csv.gz). The latest includes, for each cell, the UMAP coordinates and the retinal cell type annotation computed by Lu et al. that was used for our analysis. Normalization of the UMI counts and identification of markers for each cell type was done with Seurat as described for the retinoblastoma sample. We also looked for pan-photoreceptor markers (markers of both cones and rods). Among the markers of Cones or Rods, genes that were found overexpressed in Cones against all other types except Rods and in Rods against all other types except Cones were assigned to pan-photoreceptor. Values indicated in Supplementary Data 3 for pan-photoreceptor markers have been computed using the FindAllMarkers function comparing photoreceptor cells against all other cells.

### Visualization tool.

A R-Shiny web-app [https://retinoblastoma-retina-markers.curie.fr], based on the shiny (v.1.6.0) and shinydashboard (v.0.7.1) R-packages, was developed to visualize the expression of markers of the retina cell populations, of the two subtypes of retinoblastoma and other genes of interest cited across the manuscript in the two single-cell RNA-seq datasets (from normal human developing retina[38] and from a subtype 2 retinoblastoma, RBSC11 (this report)). The different plots and tables are made based on the R packages cowplot (v.1.1.1) and the ones included in tidyverse (v.1.3.0).

### Reporting summary.

Further information on research design is available in the Nature Research Reporting Summary linked to this article.

## Data availability

The raw array data are deposited in the Gene Expression Omnibus (GEO) database under accession code GSE58785. The raw whole-exome sequencing data are deposited in the European Genome-Pheome Archive (EGA) database under accession code EGAS00001005248. The raw targeted sequencing data are deposited in the EGA database under accession code EGAS00001005550. The raw single-cell RNA sequencing data are deposited in the EGA database under accession code EGAS00001005178. Data in EGA is available under restricted access, access can be obtained by contacting Retinoblastoma Data Access Committee – Institut Curie (data.office@curie.fr). The public retinoblastoma

transcriptomic data used in this study are available in the GEO database under accession codes GSE29683 and GSE59983. The public human developing retina scRNA-seq data used in this study are available in the GEO database under accession code GSE138002. The remaining data are available within the Article, Supplementary Information, or Source Data file. Additional data inquiry can be addressed to the Lead contact: francois.radvanyi@curie.fr. Source data are provided with this paper.

## Code availability

Codes used to generate the analysis, figures and visualization app (https://retinoblastoma-retina-markers.curie.fr) are available at Github repositories (DOI: 10.5281/zenodo.5164167, DOI: 10.5281/zenodo.5163255)[109,110].

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

## Acknowledgements

We thank Cécile Reyes and Aude Vieillefon from the Genomics platform, Benoît Albaud from the NGS platform at the Institut Curie (IC), Emmanuel Martin from Integragen for sequencing, and Nadège Gruel (IC) for her help in the single-cell experiments. We thank Dr. Jelena Martinovic and Prof. Alexandra Benachi from Béclère Hospital, Dr. M. Eugenia Riveiro from OTD Oncology, and Pr. Daniel Louvard from IC for their help with this work. Me.S. was supported by fellowships from the French MESRI and the IC, D.O. by the SFCE, the IC, and the AMCC, J.L. by the Foundation ARC, C.D. by the Barletta Foundation, Clé.H. by La Ligue Contre le Cancer. A.M.C. was funded by ISCiii-FEDER (CP13/00189). This work was supported by the Retinostop Association, the Barletta Foundation, the INCa, and the INSERM in the framework of an ICGC project

(https://icgc.org/icgc/cgp/62/355/1002881), the IC, the ICGex Equipex program, the INCa/UNADEV, the SFCE, the FES, the Association l'Etoile de Martin, the ANR in the framework of LabEx LIFESENSES (ANR-10-LABX-65), and IHU FOReSIGHT (ANR-18-IAHU-01), the XBTC sponsored by Pla Director d'Oncologia de Catalunya.

## Author contributions

Resources: L.L.L.-R., A.M., L.D., J.C.-M., H.S., H.B., F.r.D., A.M.C., N.C., G.C. and I.A. (clinical data); F.L., G.L., P.F., Ma.S., X.S.G., A.M.C. and G.C. (pathological data); Ca.D., J.C., M.G.-V., D.S.-L., L.G. and Cla.H. (genetic data); R.A., F.L., G.L., O.M. and G.P.-P. (RNA, DNA preparation). Investigation: A.N., Cé.D., C.P., D.D., D.G., D.O., I.B.-P., J.C., J.G., J.L., L.M.O., Fl.D., F.N., M.L., N.K., N.S., O.G., P.S., R.A., S.A., S.B. and S.R. Methodology: E.C., Clé.H. and C.B. (development of visualization tools). Formal analysis (bioinformatics and statistical analyses): J.L., Me.S., E.C., L.T., Clé.H., A.B., C.B., T.P., S.G., C.V., E.B., E.L., A.V. and A.d.R. Data curation: E.C. and N.E. Conceptualization: J.L., D.O., Me.S., I.B.P., S.S., X.S.-G., Fr.D., N.C., C.P., O.G., G.C., A.d.R., I.A. and F.R. Visualization: J.L., D.O., Me.S., E.C., R.A., N.S., Clé.H., F.N., E.L., C.P., G.C., A.d.R. and F.R. Writing—original draft: J.L., D.O., Me.S., G.C., A.d.R and F.R. Writing—review and editing: all authors. Funding acquisition: Me.S., Fr.D., O.G., G.C. and F.R. Supervision: A.M.C., G.C., A.d.R., I.A. and F.R. We thank the patients and their families for participating in this study.

## Competing interests

The authors declare no competing interests.

## Additional information

[1]Institut Curie, CNRS, UMR144, Equipe Labellisée Ligue contre le Cancer, PSL Research University, 75005 Paris, France. [2]Sorbonne Universités, UPMC Université Paris 06, CNRS, UMR144, 75005 Paris, France. [3]Programme Cartes d'Identité des Tumeurs, Ligue Nationale Contre le Cancer, 75013 Paris, France. [4]Precision Medicine, Hospital J.P. Garrahan, Buenos Aires, Argentina. [5]Pathology Service, Hospital J.P. Garrahan, Buenos Aires, Argentina. [6]Synergie Lyon Cancer, Plateforme de Bioinformatique "Gilles Thomas", Centre Léon Bérard, 69008 Lyon, France. [7]Département de Biologie des Tumeurs, Institut Curie, 75005 Paris, France. [8]Service de Génétique, Institut Curie, 75005 Paris, France. [9]Institut de la Vision, Sorbonne Université, INSERM, CNRS, 75012 Paris, France. [10]Institut Curie, PSL Research University, INSERM, U900, 75005 Paris, France. [11]Ecole des Mines ParisTech, 77305 Fontainebleau, France. [12]Institut Curie, CNRS, UMR3347, PSL Research University, 91405 Orsay, France. [13]Institut Curie, PSL Research University, INSERM, U1021, 91405 Orsay, France. [14]Université Paris-Saclay, 91405 Orsay, France. [15]Institut Curie, PSL Research University, INSERM U830, 75005 Paris, France. [16]Département de Recherche Translationnelle, Institut Curie, 75005 Paris, France. [17]Institut Curie, PSL Research University, NGS Platform, 75005 Paris, France. [18]GeCo Genomics Consulting, Integragen, 91000 Evry, France. [19]Département de Chirurgie, Service d'Ophtalmologie, Institut Curie, 75005 Paris, France. [20]Université de Paris, Paris, France . [21]Institut de Recerca Sant Joan de Déu, 08950 Barcelona, Spain. [22]Pediatric Hematology and Oncology, Hospital Sant Joan de Déu, 08950 Barcelona, Spain. [23]Department of Pathology, Hospital Sant Joan de Déu, 08950 Barcelona, Spain. [24]Department of Ophthalmology, Hospital Sant Joan de Déu, 08950 Barcelona, Spain. [25]National Scientific and Technical Research Council, CONICET, Buenos Aires, Argentina. [26]Département d'Imagerie Médicale, Institut Curie, 75005 Paris, France. [27]Centre de Recherche des Cordeliers, Sorbonne Universités, INSERM, 75006 Paris, France. [28]Functional Genomics of Solid Tumors, équipe labellisée Ligue Contre le Cancer, Université de Paris, Université Paris 13, Paris, France. [29]SIREDO Center (Care, Innovation and Research in Pediatric Adolescent and Young Adult Oncology), Institut Curie, 75005 Paris, France. [30]Present address: Institut Pasteur – Hub Bioinformatique et Biostatistique – C3BI, USR 3756 IP CNRS, 75015 Paris, France. [31]Present address: INSERM U930, CHU Bretonneau, 37000 Tours, France. [32]Present address: Department of Genetics, Rouen University Hospital, 76000 Rouen, France. [33]Present address: Department of Pathology, Centre Hospitalier Intercommunal de Créteil, 94000 Créteil, France. [34]These authors contributed equally: Jing Liu, Daniela Ottaviani, Meriem Sefta. [35]These authors jointly supervised this work: Guillermo Chantada, Aurélien de Reyniès, Isabelle Aerts, François Radvanyi. ✉email: francois.radvanyi@curie.fr

