## [Peer Review File · Nature Communications]

A high-risk retinoblastoma subtype with stemness features, dedifferentiated cone states and neuronal/ganglion cell gene expressionReviewers' Comments:

Reviewer #1:

Remarks to the Author:

This paper provides a thorough and helpful genomic analysis of human retinoblastoma. Based on the application of a suite of genomic analyses, the authors define two subtypes (1 and 2), both of which have RB1 loss. Subtype 1 lacks MYCN amplification, whereas Subtype 2 frequently has MYCN amplification (including a single tumor with wt RB1), and also exhibits a higher frequency of other copy number changes. Retinoblastoma is currently viewed as a tumor with few genetic lesions. This work further clarifies the disease, illustrating that subtype 2 typically has a few more copy number alterations than subtype 1. Subtype 2 also has more somatic mutations, and BCOR and ARID1 mutations, although not frequent, are unique to subtype 2. In subtype 2, tumors either have MYCNamp or, alternatively 1q gain or 16q loss, but typically not both these types of copy number events. Subtype 1 and 2 also exhibit differential methylation and gene expression patterns. Both subtypes exhibit cone gene expression, likely reflecting the cell-of-origin, but levels were considerably higher in subtype 1. Subtype 1 had two major expression clusters, exhibiting higher levels of immune response genes or metabolic genes, while subtype 2 had a higher MYC pathway profile and "stemness" genesets. By cross-comparing to data they generate from iPS-cell-derived retinas, the authors show that subtype 1 and 2 have similar levels of early cone genes, but late/more mature cone markers are reduced in subtype 2. Immunostaining confirms the two subtypes, and reveals that EBF3 is a good single marker for this purpose (off in 1, on in 2). Single cell sequencing (ssSeq), and inferred copy number variations, of a type 2 tumor suggests a two branch trajectory, both of which are CRX+, but positive or negative for EBF3. Finally, they consider metastatic retinoblastoma, and for this they turn to an additional 112 high risk cases from Buenos Aires, 19 of which included metastases. They show that TFF1 protein marks subtype 2 tumors, echoing transcriptome data, and make the interesting discovery that metastatic tumors are also TFF1 positive, suggesting that subtype 2 may be the more aggressive variety.

Overall, this is a very valuable contribution to the retinoblastoma field and will be appreciated. It is well written and, on the whole, avoids over-interpreting the data, a common flaw of genomics studies. My criticisms are not arduous and I'm highly supportive of publication with a few changes/additions.

Comments

1. The discovery of two subtypes is convincing, important and intriguing. However, the use of the term "neuronal/ganglion", particularly the focus on "ganglion" cell, to describe the gene expression pattern in subtype 2 may be misleading. Table S3 (sheet "retinal cell type markers") summarizes the genes that were manually assigned to cell type, and the references used to make these associations. This manual approach suffers the caveat that the literature may have overlooked a stage in human retinal development in a specific cell type or group of cells during which a gene, presumed to be cell type A-specific, is also involved in the development of cell type B. This problem can be minimized/overcome nowadays by correlating single cell sequencing datasets from normal development with tumor datasets. Case in point, the authors list ATOH7 as a ganglion cell gene based on data from Mellough 2019; Hoshino 2017; Kaewkhaw 2015; Wu 2015; Lu 2018. However, a ssSeq paper claims that ATOH7 is expressed in progenitor cells, amacrine-horizontal cell precursors, bipolar-photoreceptor precursors and immature photoreceptors, and provides functional evidence that it influences the cone/rod ratio (Lu et al 2020, PMID 32386599). All this to say that it seems like a good example of a gene that may not be just expressed or involved in ganglion cell development. It's possible that other markers that the authors ascribe exclusively to one cell type (e.g. "ganglion") are also not so selective. It would thus be worth comparing human retinal development ssSeq data to the retinoblastoma dataset to find out whether there is indeed a bias towards "ganglion" cell, or whether other clusters identified by ssSeq are represented. Ganglion and amacrine cells share multiple markers, including during their genesis, which is all further complicated by the genesis of multiple amacrine subtypes, some of which may be more similar to differentiating and/or mature ganglion

cells. Other markers mentioned in e.g. Fig 3E as “ganglion cell markers” are problematic, such as ONECUT1, which is also in interneurons, ELAVL3 also in amacrine cells, EBF3 also in amacrine cells etc. The authors clearly show a difference in expression levels of the indicated genes in Fig 3E in subtype 1 vs 2, which is intriguing, but the cell type labels are simplistic. Subtype 2 might be better described as “cone/neuronal” (as indeed it is finally referred to in line 560), but comparison with ssSeq clusters across developmental time will provide the best guidance.

2. A related point applies to line 457 in reference to TFF1, which shows the greatest discrimination between type 1 and 2 retinoblastoma (much higher in latter): “TFF1 is a secreted protein that is not expressed in the normal retina”. Is that still true even in ssSeq datasets of normal human retinal development? It would be useful to confirm TFF1 is truly absent throughout human retinal development, or, alternatively, discover that it is indeed present at some point in one or more lineage.

3. The authors note that there are no notable gene expression differences between MYCN amplified and non-MYCN-amplified tumors (e.g. line 540, Discussion). In that case, it must also include MYCN, implying that whether the MYCN gene is amplified or not, MYCN transcript is expressed at similar levels. The authors should discuss this point and, preferably, show a scatter plot with MYCN transcript levels in the MYCN amplified or non-MYCN amplified subsets, and test whether, as predicted, there is no significant difference.

4. Bottom of page 15 – the authors argue that the ARR3+/EBF3- regions that are mostly Ki-67-negative might correspond to retinoma because they have (mostly) stopped proliferating. This seems to be a misunderstanding of tumor homeostasis; a tumor is like any tissue (albeit disorganized), and consists of a range of cell type stages, from stem to progenitor to differentiating to terminally differentiated, with many of the latter being post-mitotic. So, naturally, not every tumor cell is dividing. This is the case across cancer (the Ki67 index is not 100%). The ARR3+/EBF3- cells could be at a later stage of differentiation than the ARR3-/EBF3+ cells, and is thus more likely to have exited the cell cycle. The latter seems more likely than the retinoma scenario, as these benign tumors are small, and only detected in a small fraction (~15%) of retinoblastoma cases. The description of post-mitotic photoreceptor fleurettes in areas of the tumor are consistent with terminal differentiation. Cell-cycle exit associated with terminal differentiation seems the most probable explanation for Ki67-negative ARR3+/EBF3- cells. The authors should state both possibilities.

5. A caveat with the ssSeq data is it's one sample. I don't expect the authors to perform additional ssSeq, but they should note this caveat in the Results section.

6. The authors should clarify in the text (e.g. as rationale for ssSeq analysis) that there are distinct explanations for bulk analysis expression results showing different lineage markers in the same tumor. In one case, it may reflect the presence of different lineages derived from a common precursor, or it could reflect aberrant co-expression in tumor cells of genes that are normally expressed in distinct cell types. Both scenarios are possible within the same tumor. The only way to know is to compare that tumor ssSeq data to ssSeq data from normal human retinal development. It would be very helpful if the authors performed such an analysis. It reveals, for example, whether the CRX+/EBF3+/GAP43+ cells observed in the subtype 2 tumors subject to ssSeq (Fig 4e) are unique to retinoblastoma, or whether they are also a normal feature of retinal development. The authors imply in the text that EBF3 and GAP43 expression show ganglion cell enrichment, but the literature implies a role for EBF3 in the amacrine lineage (at least in mouse, PMID 20826655, 29258872), Table S3 acknowledged that dual expression pattern. However, although Table S3 assigns GAP43 solely to ganglion cells, it is very likely expressed in the amacrine lineage. CRX in Table S3 is assigned to photoreceptors, but that is also not the case as it's expressed in bipolar cells too. During human retinal development, is it possible that CRX, EBF3 and GAP43 are co-expressed in a subset of differentiating neurons? Or is this combination unique to tumor cells? Past papers have made the latter type of claim, but without the

benefit of ssSeq analysis of normal human retinal development. Such data is now available to better address the two contrasting models, and thus (finally) to explain unanticipated co-expression patterns in tumor cells.

7. This next suggestion is not essential, but to complement the lineage diagram in Fig 5e, which is based on copy number changes in clusters inferred from ssSeq expression data, it would be interesting to test the lineage relationship of cells in the ssSeq data using an approach such as RNA velocity. Again, this is not essential for the paper, but if the authors don't perform the analysis, they could at least suggest that such work could test the suggested lineage relationships in 5e.

8. The metastatic analysis is fascinating, but the conclusion is based on analysis of a single marker (TFF1). The authors should clearly state that caveat when they make the claim that subtype tumors are more likely to metastasize e.g. they could state that this result needs to be confirmed with other subtype 2 features that they identified in their paper.

Minor point

Fig 2d – Re the colors used to signify RB1 mutation type - It's very difficult to tell the difference between the colors for a nonsense mutation or promoter methylation. Could the authors change the code to improve?

Reviewer #2:

Remarks to the Author:

The study used transcriptome, methylome and somatic DNA copy number alteration data from 102 retinoblastoma to identify two retinoblastoma subtypes. Detailed analysis with immunohistochemistry, organoids and scRNAseq, the authors investigated different molecular and histopathological features between these two subtypes. They suggested that subtype 2 tumors were less differentiated and had higher metastasis potential than subtype 1. While the analysis of multiple types of omics data in human samples and in vitro models were comprehensive and produced valuable hypotheses, the conclusions (e.g. the two cancer subtypes and their different features) need to be confirmed by additional data and analyses.

Major comments:

- The molecular signatures that distinguish the two subtypes need to be validated in independent patient cohort(s). It is unclear how the classification of the two subtypes can be applied for classifying unseen data. The authors did use a second cohort, with 112 primary tumors (19 of which had metastasis). However, only one marker TFF1 was used as the marker for distinguishing the subtype 2 from subtype 1. However, this single-marker approach does not reflect the multi-marker classification, a main result from most of the analyses reported in the manuscript. The use of organoids from a human iPSC cell line would not represent genetic variation between different patients as in a patient cohort.
- It was unclear how the clustering threshold (resolution) was used to define the number of clusters as two clusters. With different clustering resolutions, there could be more than two subtypes. For example, the gene clustering result as shown in Fig. 3b suggested a gene group (Gene cluster 1.1.) could split the sample subtype 1 into two subsets of samples. One subset appeared to have high stemness indices (Fig. 3d). It is, therefore, possible to exist three subtypes.

Minor comments:

- Line 155: briefly introduce consensus clustering when the concept first appears in the manuscript. This concept can be understood in many senses. Similarly, the cluster-of-clusters approach seems to

be a new name/concept and should be described briefly in the Result section before going into details of the clustering results.

- Unsupervised and supervised centroid-based clustering are general terminology and need to be more specific. It would be clearer to distinguish/relate them to the traditional hierarchical clustering and kmeans clustering?

- Line 166: the 9-CpG classifier wasn't described clearly in the Methods section. It looks like the authors selected 9 significantly different CpG methylation to form a classifier, but how the classifier was trained and formulated? Also, the name 9-CpG is unconventional and can cause confusion with the naming system for 5' CpG methylation

- Line 421 "whereas clusters 5 and 6 corresponded to normal cells" -> the cluster 5 looks like it has strong CNV on chromosome 6.

- Methods for generating organoids were not described

- I would recommend that the codes should be made publicly available for reproducibility and broader applications because this is an analysis-based manuscript

We thank the reviewers for their constructive comments and suggestions, which have enabled us to substantially improve this manuscript.

Reviewer # 1

This paper provides a thorough and helpful genomic analysis of human retinoblastoma. Based on the application of a suite of genomic analyses, the authors define two subtypes (1 and 2), both of which have RB1 loss. Subtype 1 lacks MYCN amplification, whereas Subtype 2 frequently has MYCN amplification (including a single tumor with wt RB1), and also exhibits a higher frequency of other copy number changes. Retinoblastoma is currently viewed as a tumor with few genetic lesions. This work further clarifies the disease, illustrating that subtype 2 typically has a few more copy number alterations than subtype 1. Subtype 2 also has more somatic mutations, and BCOR and ARID1 mutations, although not frequent, are unique to subtype 2. In subtype 2, tumors either have MYCNamp or, alternatively 1q gain or 16q loss, but typically not both these types of copy number events. Subtype 1 and 2 also exhibit differential methylation and gene expression patterns. Both subtypes exhibit cone gene expression, likely reflecting the cell-of-origin, but levels were considerably higher in subtype 1. Subtype 1 had two major expression clusters, exhibiting higher levels of immune response genes or metabolic genes, while subtype 2 had a higher MYC pathway profile and “stemness” genesets. By cross-comparing to data they generate from iPS-cell-derived retinas, the authors show that subtype 1 and 2 have similar levels of early cone genes, but late/more mature cone markers are reduced in subtype 2. Immunostaining confirms the two subtypes, and reveals that EBF3 is a good single marker for this purpose (off in 1, on in 2). Single cell sequencing (ssSeq), and inferred copy number variations, of a type 2 tumor suggests a two branch trajectory, both of which are CRX+, but positive or negative for EBF3. Finally, they consider metastatic retinoblastoma, and for this they turn to an additional 112 high risk cases from Buenos Aires, 19 of which included metastases. They show that TFF1 protein marks subtype 2 tumors, echoing transcriptome data, and make the interesting discovery that metastatic tumors are also TFF1 positive, suggesting that subtype 2 may be the more aggressive variety.

Overall, this is a very valuable contribution to the retinoblastoma field and will be appreciated. It is well written and, on the whole, avoids over-interpreting the data, a common flaw of genomics studies. My criticisms are not arduous and I'm highly supportive of publication with a few changes/additions.

Comments

1.1 The discovery of two subtypes is convincing, important and intriguing. However, the use of the term “neuronal/ganglion”, particularly the focus on “ganglion” cell, to describe the gene expression pattern in subtype 2 may be misleading. Table S3 (sheet “retinal cell type markers”) summarizes the genes that were manually assigned to cell type, and the references used to make these associations. This manual approach suffers the caveat that the literature may have overlooked a stage in human retinal development in a specific cell type or group of cells during which a gene, presumed to be cell type A-specific, is also involved in the development of cell type B. This problem can be minimized/overcome nowadays by correlating single cell sequencing datasets from normal development with tumor datasets. Case in point, the authors list ATOH7 as a ganglion cell gene based on data from Mellough 2019; Hoshino 2017; Kaewkhaw 2015; Wu 2015; Lu 2018. However, a ssSeq paper claims that ATOH7 is expressed in progenitor cells, amacrine-horizontal cell precursors, bipolar-photoreceptor precursors and

immature photoreceptors, and provides functional evidence that it influences the cone/rod ratio (Lu et al 2020, PMID 32386599). All this to say that it seems like a good example of a gene that may not be just expressed or involved in ganglion cell development. It's possible that other markers that the authors ascribe exclusively to one cell type (e.g. "ganglion") are also not so selective. It would thus be worth comparing human retinal development ssSeq data to the retinoblastoma dataset to find out whether there is indeed a bias towards "ganglion" cell, or whether other clusters identified by ssSeq are represented. Ganglion and amacrine cells, share multiple markers, including during their genesis, which is all further complicated by the genesis of multiple amacrine subtypes, some of which may be more similar to differentiating and/or mature ganglion cells. Other markers mentioned in e.g. Fig 3E as "ganglion cell markers" are problematic, such as ONECUT1, which is also in interneurons, ELAVL3 also in amacrine cells, EBF3 also in amacrine cells etc. The authors clearly show a difference in expression levels of the indicated genes in Fig 3E in subtype 1 vs 2, which is intriguing, but the cell type labels are simplistic.

We agree with the reviewer that, for several markers, stage(s) in human retinal development in specific cell types can have been overlooked. Indeed, it was particularly important, for the selection of retinal cell-type markers in this study, to take into account recently published single-cell RNAseq data of the developing normal human retina by Lu *et al.* ³⁸.

We first identified lists of candidate markers associated with each retinal cell type from the annotated cell types defined by Lu *et al.* For this purpose, we used the "FindAllMarkers" function of Seurat. These lists, ranked according to the log fold-change of the average expression between the type of interest and all the other cells (avg_logFC), are provided in an additional tab of Supplementary Table 3 "Cell type markers_Lu data".

In order to choose the most specific cell-type markers, we developed a tool for visualizing gene expression profiles in the different retinal cell types at different time points during development. This tool is described in the Methods section and a link to it is provided in the manuscript (<https://retinoblastoma-retina-markers.curie.fr/>).

The analysis of the expression profiles of the candidate markers obtained from Lu *et al.* data and of markers found in the literature led us to propose markers for the different retinal cell types. The list of the most specific cell-type markers is provided in tab "Selected cell type markers" of Supplementary Table 3.

Figure 3e and Extended data Fig. 4 have been modified in accordance with this updated version of the list of cell type-specific markers.

The results on the expression of cell type markers in the two subtypes still hold. Both subtypes expressed cone photoreceptor markers with down regulation of late cone markers in subtype 2. A set of retinal ganglion cell genes was overexpressed in subtype 2 tumors. Most of the markers of other retinal cell types (rods, amacrine, bipolar, horizontal, and Müller glia cells) were not expressed in retinoblastomas or were only expressed in a subset of tumors, probably due to the

presence of normal cells in these tumors. Indeed normal retinal cells have been shown to be present in some retinoblastomas¹³.

For an unbiased analysis of the potential enrichment in markers of particular retinal cell types among the genes overexpressed in subtype 2 tumors, we performed an enrichment analysis with the top retinal markers automatically retrieved from Lu *et al.*³⁸ Using these ranked lists, we found that the retinal cell-type markers displaying the highest degree of enrichment among the genes overexpressed in subtype 2 tumors were those associated with retinal ganglion cells ($p=2.9 \cdot 10^{-8}$), followed by markers associated with precursor of amacrine/horizontal cells ($p=3.4 \cdot 10^{-2}$). This new analysis is described in the Results section (*page 12-13, line 313-322*) and the results given in an additional tab “RetinaMarker enrichment gcluster” of Supplementary Table 3.

1.2 Subtype 2 might be better described as “cone/neuronal” (as indeed it is finally referred to in line 560), but comparison with ssSeq clusters across developmental time will provide the best guidance.

In the previous version of the manuscript, we reported neuronal/ganglion cell gene expression as one of the features of subtype 2 tumors. Using the refined list of specific ganglion markers, considering Lu *et al.*'s single-cell data, we still observe the upregulation of a subset of ganglion cell markers in subtype 2 tumors. Moreover, the genes upregulated in subtype 2 were also found to be enriched in the ganglion cell marker list automatically retrieved from Lu *et al.*'s data. For these reasons, it seems justified to keep the term “ganglion”. However, the ganglion cell genes overexpressed in subtype 2, although specific to this cell type in the context of the retina, are also expressed in the brain and play various functions in the central nervous system. In addition, several neuronal genes not specific to ganglion cells (also found in other retinal cell types) are overexpressed in subtype 2. We therefore agree with the reviewer that subtype 2 may be considered as a cone-neuronal subtype.

We now discuss the terms “ganglion” and “neuronal” in the Results (*page 13, line 326-335*) and Discussion (*page 23, line 577-593*).

2. A related point applies to line 457 in reference to TFF1, which shows the greatest discrimination between type 1 and 2 retinoblastoma (much higher in latter): “TFF1 is a secreted protein that is not expressed in the normal retina”. Is that still true even in ssSeq datasets of normal human retinal development? It would be useful to confirm TFF1 is truly

absent throughout human retinal development, or, alternatively, discover that it is indeed present at some point in one or more lineage.

Analysis of scRNA-seq data confirmed that *TFF1* was expressed very little, if at all, during normal retina development. A supplementary figure has been added: Extended Data Fig. 6a. Interestingly, most of the few cells expressing *TFF1* corresponded to cone cells at early stages of development, suggesting that there may be a permissive state for *TFF1* expression early in cone differentiation. Additional data (single-cell and *in situ* analyses) will be required to confirm this.

3. The authors note that there are no notable gene expression differences between MYCN amplified and non-MYCN-amplified tumors (e.g. line 540, Discussion). In that case, it must also include MYCN, implying that whether the MYCN gene is amplified or not, MYCN transcript is expressed at similar levels. The authors should discuss this point and, preferably, show a scatter plot with MYCN transcript levels in the MYCN amplified or non-MYCN amplified subsets, and test whether, as predicted, there is no significant difference.

Our sentence in the Discussion “Although no genes were found to be significantly differentially expressed between *MYCN*-non amplified and *MYCN*-amplified within subtype 2 tumors” is incorrect and we apologize for this error. There were much less genes significantly differentially expressed between subtype 2 tumors with and without *MYCN* amplification than between subtype 1 and subtype 2 tumors, but there were nevertheless some such genes, including *MYCN* and the neighboring genes. We have corrected this (*page 25, line 611-612*) as follows: “*MYCN*-amplified tumors did not cluster separately from other subtype 2 tumors on transcriptome analyses, but they nevertheless had specific features”.

4. Bottom of page 15 – the authors argue that the ARR3+/EBF3- regions that are mostly Ki-67-negative might correspond to retinoma because they have (mostly) stopped proliferating. This seems to be a misunderstanding of tumor homeostasis; a tumor is like any tissue (albeit disorganized), and consists of a range of cell type stages, from stem to progenitor to differentiating to terminally differentiated, with many of the latter being post-mitotic. So, naturally, not every tumor cell is dividing. This is the case across cancer (the Ki67 index is not 100%). The ARR3+/EBF3- cells could be at a later stage of differentiation than the ARR3-/EBF3+ cells, and is thus more likely to have exited the cell cycle. The latter seems more likely than the retinoma scenario, as these benign tumors are small, and only detected in a small fraction (~15%) of retinoblastoma cases. The description of post-mitotic photoreceptor fleurettes in areas of the tumor are consistent with terminal differentiation. Cell-cycle exit associated with terminal differentiation seems the most probable explanation for Ki67-negative ARR3+/EBF3- cells. The authors should state both possibilities.

We have now inserted these two possibilities in the text (*page 17, line 420-425*), referring to the $ARR3^-/EBF3^+$ and non-proliferating $ARR3^+/EBF3^-$ areas:

“The presence of these different areas within the tumor could reflect a range of tumor cell type stages, from stem, to progenitor to differentiating to terminally differentiated, with many of the latter being post-mitotic. Alternatively, the Ki-67-negative $ARR3^+/EBF3^-$ areas could correspond to retinoma, a benign non-proliferative lesion observed adjacent to retinoblastoma.” In favor of this second possibility, the size of retinoma reported in the literature (Margo *et al.*, 1983, PMID: 6626001) is similar to the size of the $ARR3^+/EBF3^-$ areas observed in the subtype 2 tumors with mutually exclusive *ARR3* and *EBF3* expression. In addition, the percentage of retinoma reported in retinoblastoma (~15%) is not incompatible with the number of subtype 2 tumors with a mutually exclusive expression pattern (30% of subtype 2 tumors). These arguments have not been included in the manuscript, as we are awaiting the results of analyses for additional tumors.

5. A caveat with the ssSeq data is it's one sample. I don't expect the authors to perform additional ssSeq, but they should note this caveat in the Results section.

We have noted this caveat in the Results section (*page 19, line 484-488*):

“The single-cell RNA-seq analysis was performed on only one retinoblastoma. Single cell analysis of additional tumors of both subtypes are necessary to further assess retinoblastoma heterogeneity and to investigate the relationship between retina development and tumorigenesis...”

*6. The authors should clarify in the text (e.g. as rationale for ssSeq analysis) that there are distinct explanations for bulk analysis expression results showing different lineage markers in the same tumor. In one case, it may reflect the presence of different lineages derived from a common precursor, or it could reflect aberrant co-expression in tumor cells of genes that are normally expressed in distinct cell types. Both scenarios are possible within the same tumor. The only way to know is to compare that tumor ssSeq data to ssSeq data from normal human retinal development. It would be very helpful if the authors performed such an analysis. It reveals, for example, whether the $CRX^+/EBF3^+/GAP43^+$ cells observed in the subtype 2 tumors subject to ssSeq (Fig 4e) are unique to retinoblastoma, or whether they are also a normal feature of retinal development. The authors imply in the text that *EBF3* and *GAP43* expression show ganglion cell enrichment, but the literature implies a role for *EBF3* in the amacrine lineage (at least in mouse, PMID 20826655, 29258872), Table S3 acknowledged that dual expression pattern. However, although Table S3 assigns *GAP43* solely to ganglion cells, it is very likely expressed in the amacrine lineage. *CRX* in Table S3 is assigned to photoreceptors, but that is also not the case as it's expressed in bipolar cells too. During human retinal development, is it possible that *CRX*, *EBF3* and *GAP43* are co-expressed in a subset of*

differentiating neurons? Or is this combination unique to tumor cells? Past papers have made the latter type of claim, but without the benefit of ssSeq analysis of normal human retinal development. Such data is now available to better address the two contrasting models, and thus (finally) to explain unanticipated co-expression patterns in tumor cells.

Despite the dropout effect of scRNAseq, meaning that only a small fraction of the transcriptome of each cell is captured, a substantial proportion of the early cone-neuronal/ganglion tumor cells coexpressed an early cone marker (CRX) and neuronal/ganglion cell markers (EBF3, GAP43). This combination was specific to the tumor cells because this coexpression of cone and neuronal/ganglion cell markers was almost never observed during retinal development (only 56 of 118 555 cells).

We have added one Figure in the Extended data to show this co-expression in tumor cells of a subtype 2 tumor and its absence in normal retinal cells during development (Extended data Fig.5f) and added in the Results section (*page 19, line 481-483*) “The co-expression of *CRX/EBF3/GAP43* (early photoreceptor/cone marker and neuronal/ganglion cell markers) observed in tumor cells was absent or very rare during normal retinal development (Extended Data Fig. 5f)”.

We have included in our visualization tool the possibility to visualize co-expression of different retinal markers and of genes differentially expressed between the two subtypes (see Gene Co-expression at <https://retinoblastoma-retina-markers.curie.fr>). We used this tool to generate the Extended Data Fig. 5f.

7. This next suggestion is not essential, but to complement the lineage diagram in Fig 5e, which is based on copy number changes in clusters inferred from ssSeq expression data, it would be interesting to test the lineage relationship of cells in the ssSeq data using an approach such as RNA velocity. Again, this is not essential for the paper, but if the authors don't perform the analysis, they could at least suggest that such work could test the suggested lineage relationships in 5e.

We have performed analyses based on RNA velocity with publicly available tools (loompy, scVelo). However, in our case, cell cycle was a major confounding factor (see Figure below, the arrows point to cycling cells). It is reported that such algorithms need sufficient sampling of initial and transitional cells to properly infer lineage relationships between multiple tumor subpopulations⁶⁰. The use of these methods in retinoblastoma will, therefore, require advanced preprocessing steps, finer parameter tuning and, most importantly, additional samples like in Couturier et al., 2020 (PMID: 32641768) where 16 glioblastoma samples were analyzed.

As suggested by the reviewer, we mentioned the use of these methods as a future perspective in the context of analysis of additional retinoblastoma scRNA-seq data: “Single cell analysis of additional tumors of both subtypes are necessary to further assess retinoblastoma heterogeneity and to investigate the relationship between retinal development and tumorigenesis using trajectory inference methods such as the ones estimating RNA velocity^{59, 60}”.

8. *The metastatic analysis is fascinating, but the conclusion is based on analysis of a single marker (TFF1). The authors should clearly state that caveat when they make the claim that subtype tumors are more likely to metastasize e.g. they could state that this result needs to be confirmed with other subtype 2 features that they identified in their paper.*

We clearly state this caveat in the Results section (*page 21, line 531-533*): “These findings require validation by additional evidence for subtype 2 assignment, and by studies on additional series of metastatic cases”.

We have also strengthened our results by analyzing the expression of EBF3, a ganglion cell marker, which was among the most upregulated genes in subtype 2 tumors (Fig. 3a). It was expressed in most subtype 2 tumors with little or no expression in subtype 1 tumors, as demonstrated by analyses of both RNA and protein levels (Fig. 4d,e, Extended Data Fig. 6). Expression could be determined by immunohistochemistry in 16 of 19 tumors from which metastases subsequently developed. All but one of the 16 samples analyzed were positive for EBF3 (QS > 270). Six metastatic sites were also analyzed, and all were positive for EBF3 (QS > 255). These results are now reported in the manuscript (*page 21, line 523-526, 528-529*) and in Supplementary Table 6.

Further evidence supporting the use of TFF1 and EBF3 as subtype 2 markers was provided by the analyses of two additional retinoblastoma transcriptome series (analyses performed in response to one of the comments from reviewer 2). Indeed, in these two additional series, as in our series, TFF1 and EBF3 were among the most significantly differentially expressed genes in subtype 2 tumors. An additional figure has been included which presents these results from these two additional series (Extended Data Fig. 7).

Minor point

Fig 2d – Re the colors used to signify RB1 mutation type - It's very difficult to tell the difference between the colors for a nonsense mutation or promoter methylation. Could the authors change the code to improve?

The colors for a nonsense mutation and promoter methylation were indeed very similar. We have changed the colors to make it easier to distinguish between these two categories.

Reviewer # 2

The study used transcriptome, methylome and somatic DNA copy number alteration data from 102 retinoblastoma to identify two retinoblastoma subtypes. Detailed analysis with immunohistochemistry, organoids and scRNAseq, the authors investigated different molecular and histopathological features between these two subtypes. They suggested that subtype 2 tumors were less differentiated and had higher metastasis potential than subtype 1. While the analysis of multiple types of omics data in human samples and in vitro models were comprehensive and produced valuable hypotheses, the conclusions (e.g. the two cancer subtypes and their different features) need to be confirmed by additional data and analyses.

Major comments:

- The molecular signatures that distinguish the two subtypes need to be validated in independent patient cohort(s).

We took advantage of two publicly available retinoblastoma patient cohorts with tumor transcriptome data^{16,18} (GSE29683, GSE59983) for 55 and 76 patients, respectively. Partial clinical data were available (age at diagnosis (rank) and heredity for all patients in Kooi's data and age at diagnosis and heredity for 29 patients for McEvoy's data). 46 of 55 tumors in McEvoy *et al.* and 74 of 76 tumors in Kooi *et al.* could be assigned to either of the two subtypes using nearest centroid classification with the mRNA predictor obtained from our series.

We were able to validate the molecular features based on transcriptome analyses that distinguish the two subtypes in these two independent series: higher stemness signature, and E2F and MYC target gene meta-scores in subtype 2, higher immune cell abundance and interferon response pathway meta-score in subtype 1. We could also validate that the two subtypes differed in clinical features: age at diagnosis differed significantly between the two subtypes in the two additional series, heredity was also significantly different between the two subtypes in the Kooi series, and the same trend for this parameter was observed in the McEvoy series.

Concerning retinal differentiation assessed by transcriptomic analysis, the same results were obtained as in our cohort: a similar expression of early cone markers in both subtypes, with a downregulation of the expression of late cone markers in subtype 2, and a significant overexpression of retinal ganglion cell markers in subtype 2.

These results have been added in the Results (*page 20, line 511-512*) and Discussion (*page 22, line 540-541*). They are included in two supplementary figures (Extended data Fig. 6 and 7) and two tabs (limma (Kooi et al.), limma (McEvoy et al.)) in Supplementary Table 3. A paragraph describing the methods used has been added to the Methods section (*page 52, line 1261-1278*).

- *It is unclear how the classification of the two subtypes can be applied for classifying unseen data. The authors did use a second cohort, with 112 primary tumors (19 of which had metastasis). However, only one marker TFF1 was used as the marker for distinguishing the subtype 2 from subtype 1. However, this single-marker approach does not reflect the multi-marker classification, a main result from most of the analyses reported in the manuscript.*

This series of 112 primary tumors with high-risk pathological features (Garrahan series) was particularly valuable, however a limited amount of material was available. In addition to TFF1, we performed immunohistochemical analysis for EBF3, a ganglion/neuronal cell marker found, like TFF1, to be one of the most significantly overexpressed genes in subtype 2 tumors in our initial cohort (Fig. 3a). EBF3 was expressed in most subtype 2 tumors with little or no expression in subtype 1 tumors, as assessed at both protein and RNA levels (Fig. 4d,e, Extended Data Fig. 6). Sixteen of the 19 cases that went on to develop metastases were evaluable for this EBF3 immunohistochemical analysis, and 15 of these 16 cases had a high quick score (>270) for EBF3. Six metastatic sites could also be analyzed, and all were positive for EBF3 (QS > 255). These immunohistochemistry results for EBF3 have been added to Supplementary Table 6.

We also confirmed in the two independent cohorts from Kooi et al.¹⁸ and McEvoy et al.¹⁶, that *TFF1* and *EBF3* were, as in our initial series, among the genes most significantly overexpressed in subtype 2, and that they were expressed in most subtype 2 tumors and not or little expressed in subtype 1 tumors. An additional figure including these results have been added (Extended Data Fig. 6).

However our study has still limitations and we acknowledge it in the Results section (*page 21, line 531-533*) by writing “These findings require validation by additional evidence for subtype 2 assignment, and by studies on additional series of metastatic cases”.

- *The use of organoids from a human iPSC cell line would not represent genetic variation between different patients as in a patient cohort.*

We agree with the referee that culture conditions and induction of differentiation conditions are not the sole source of variation in iPSC studies and that genetic variability can be a major source of variation.

We previously demonstrated the efficacy of our retinal differentiation protocol with different iPS cell lines derived from adult dermal fibroblasts⁵², neonatal foreskin fibroblasts⁴⁸ and retinal Müller glial cells⁷⁷.

In the experiments using iPS cells reported here, we used only one cell line (hiPSC-2 clone⁵²). However the time of appearance of early and late cone markers were found to be very similar in different iPS cell lines from different sources and different donors, and in the normal developing retina (results obtained from the single cell data from Lu et al. (2020)). We have added more detail to the methods, together with the clone used:

“By RT-qPCR and immunofluorescence analysis, we previously showed that the different iPS cell lines (hiPSC-2 clone⁵², AAVS1:CrXP_H2BmCherry hiPSC line⁷⁷) we used are able to produce the whole repertoire of retinal cells, in an identical way and following the same chronological order with first the appearance of ganglion cells, then the amacrine and horizontal cells and finally the mature photoreceptors, the bipolar cells and the Müller glial cells. The use of different markers of photoreceptor lineage (CRX, RCVRN, NRL, NR2E3, ARR3, RHO, OPSINS...) showed that the genesis of cones and rods is identical in the different iPS cell lines used”.

- It was unclear how the clustering threshold (resolution) was used to define the number of clusters as two clusters. With different clustering resolutions, there could be more than two subtypes. For example, the gene clustering result as shown in Fig. 3b suggested a gene group (Gene cluster 1.1.) could split the sample subtype 1 into two subsets of samples. One subset appeared to have high stemness indices (Fig. 3d). It is, therefore, possible to exist three subtypes.

Here we first performed consensus clustering both within each omics and for different values of the number k of clusters (k from 2 to 8). Given an omics (ex. transcriptome) and a number k of clusters (ex. $k=5$), we performed 24 hierarchical clustering using 8 data subsets (with varying number of highest variable features), Pearson's metrics and 3 different linkage methods. Then we calculated the consensus partition in k clusters across those 24 partitions (using the co-classification matrix built from these 24 partitions). We observed both for transcriptome and methylome that the (intra-omics) consensus partition with $k=2$ clusters was more stable than solutions with $k>2$ clusters. We thus selected $k=2$ clusters for these two omics. The DNA copy number data yielded 5 clusters. Then the multi-omics cluster-of-cluster analysis finally yielded 2 clusters, largely driven by a major convergence between the transcriptome and methylome partitions in 2 clusters. At the end, we claim that there are at least two main subtypes that have relevant clinical and molecular differences. We agree with the reviewer that there could be more than 2 subtypes. One must say that it is difficult to fully avoid subjectivity when performing unsupervised class discovery: metrics exist to look for the "best" number k of clusters, but are

intrinsically questionable since they all depend on k. With further refinements and integration of additional samples, one could perhaps find additional subgroups.

Details of the method used are provided in the Methods section (*page 45-47, line 1098-1132*).

Minor comments:

- Line 155: briefly introduce consensus clustering when the concept first appears in the manuscript. This concept can be understood in many senses.

Several approaches to consensus clustering exist; they all aim to derive a consensus partition (or tree) from several initial partitions (or trees) of a set of objects. In the frame of omics studies, consensus partitions can be derived either from intra-omics analyses or from multiple omics analyses. In our manuscript, we both used intra-omics consensus clustering and multiple omics consensus clustering, the latter being called cluster-of-clusters analysis to distinguish it from the former. In both cases we used the same approach to derive a consensus partition from a series of initial partitions: briefly, we build a (samples x samples) co-classification matrix from the set of initial partitions; this co-classification (similarity) matrix was transformed into a matrix of dissimilarity which was then used to perform hierarchical clustering of the samples using complete linkage, and the consensus partition was finally extracted from the obtained dendrogram using a cut in k groups. To clarify this point we modified the manuscript (*page 6, line 154-161*) as follows: “Within each of these three omics datasets, we calculated several partitions of the samples in k clusters (k-partitions), for various values of k, through unsupervised hierarchical clustering, using varying numbers of features and different linkages (see Methods). Then, for each omics and each value of k, we performed a consensus clustering analysis to derive a consensus k-partition. Doing so the transcriptome-based and methylome-based analyses both yielded stable consensus partitions in two clusters, while the SCNA-based analysis yielded a stable consensus partition in five clusters (Fig. 1a, upper panel, Extended Data Fig. 1a). Cluster memberships from each of the three partitions were analyzed by a cluster-of-clusters approach, briefly, a sample co-classification matrix was built and was then subjected to hierarchical clustering using complete linkage.”.

- Similarly, the cluster-of-clusters approach seems to be a new name/concept and should be described briefly in the Result section before going into details of the clustering results.

As mentioned above, we call “cluster-of-clusters” analysis a consensus clustering analysis applied to partitions derived from multiple omics. To clarify this point we modified the manuscript as mentioned above.

- Unsupervised and supervised centroid-based clustering are general terminology and need to be more specific. It would be clearer to distinguish/relate them to the traditional hierarchical clustering and kmeans clustering?

Unsupervised clustering analysis is now described in more details (see above). Regarding supervised prediction using a centroid-based approach, we modified the text in the Methods (page 47, line 1134-1164), indicating the way centroids are obtained, and the nearest centroid metrics used to predict the class of a sample: “To classify these remaining samples according to either subtype 1 or subtype 2, we built two supervised centroid-based predictors, one transcriptomic and the other methylomic. The two core clusters defining subtype 1 and 2 were used to train these classifiers. For the transcriptomic data, the centroids of subtype 1 and subtype 2 were calculated as the intra-cluster median expression of the 800 genes most significantly differentially expressed between the two clusters (taking the 400 most upregulated genes in each subtype); similarly, for the methylomic data, the centroids of subtypes 1 and 2 were based on the median beta value of the 10000 CpGs most significantly differentially methylated between the two clusters (5000 most methylated in each subtype).”.

- Line 166: the 9-CpG classifier wasn't described clearly in the Methods section. It looks like the authors selected 9 significantly different CpG methylation to form a classifier, but how the classifier was trained and formulated? Also, the name 9-CpG is unconventional and can cause confusion with the naming system for 5' CpG methylation.

We now describe the nine-CpG-based classifier in more detail in the Methods (paragraph “Array-based methylation signature”):

From the methylome array data (n=66), we selected the most differentially methylated CpGs between the two retinoblastoma subtypes (clusters of clusters) based on statistics of Wilcoxon test. Out of top 50 hypermethylated CpGs and top 30 hypomethylated CpGs of subtype 2 retinoblastoma (by *p*-value), top 7 hypermethylated and top 7 hypomethylated CpGs by difference of beta value were selected for pyrosequencing. 5 of them didn't perform well in pyrosequencing. This method led to the selection of 9 CpGs significantly differentially methylated that have been analyzed by pyrosequencing for sample classification. Seventeen samples from the initial series were analyzed by pyrosequencing for validation of the nine-CpG-based-classifier (9 subtype 1, 8 subtype 2); from these samples we derived a subtype 1 and subtype 2 centroids based on these 9 CpGs. The nearest-centroid approach (with Pearson's metric and a minimal threshold of 0.3) correctly assigned 16 of these 17 samples to their known subtype and left unassigned the remaining sample. Additional samples analyzed by

pyrosequencing for these 9 CpGs were then classified using the nearest-centroid approach (Pearson's metric) at a minimal threshold of 0.3.

We have also changed the name of the 9-CpG classifier to the "nine-CpG-based classifier" throughout the manuscript.

- Line 421 "whereas clusters 5 and 6 corresponded to normal cells" -> the cluster 5 looks like it has strong CNV on chromosome 6.

CNV has been inferred from the transcriptomic data, assuming that CNV induces the simultaneous upregulation or downregulation of neighboring genes. More rarely, neighboring genes may be simultaneously up- or downregulated independently of copy number, as we and others have reported (Stransky et al., 2006, PMID: 17099711). Based on the genes expressed, the cluster 5 cell population very likely corresponds to macrophages/microglia. This cell population overexpresses HLA genes, which are clustered on chromosome 6. The clustering and overexpression of these genes probably account for the band observed on chromosome 6 on Figure 5d for this cell population. We now state in the legend to Figure 5: "The genes on chromosome 6p overexpressed in non-malignant monocyte/microglia cells correspond to HLA complex genes and should not be interpreted as CNV in cluster 5".

- Methods for generating organoids were not described

We describe the methods used to generate organoids in the Methods, in the paragraph "Human iPSC maintenance and retinal organoid generation", specifying the cell line used:

"Human induced pluripotent stem cells (iPSCs) derived from dermal fibroblasts (hiPSC-2 clone)⁵² were cultured on truncated recombinant human vitronectin-coated dishes with Essential 8TM medium (ThermoFisher Scientific), as previously described⁴⁸. For retinal differentiation, adherent iPSCs were expanded to 70-80%, then FGF-free medium was added to the cultures for 2 days followed by a neural induction period allowing the appearance of retinal structures. Identified retinal organoids were manually isolated around day 28 and cultured as floating structures for the next several weeks to follow retinal differentiation and maturation as previously described^{48,76}".

- I would recommend that the codes should be made publicly available for reproducibility and broader applications because this is an analysis-based manuscript

The codes and raw data have been made publicly available (see page 56, line 1360-1366), as requested by the reviewer and the editor.

Reviewers' Comments:

Reviewer #1:

Remarks to the Author:

The authors have thoroughly addressed my concerns. I thank them for their work in clarifying the points raised. I strongly support publication of their work.

Reviewer #3:

Remarks to the Author:

This is a well done study that describes two subtypes of retinoblastoma. The authors properly addressed the reviewers' comments and revised the manuscript accordingly. I have one minor comment that was not raised in the previous round. Could the authors regress out cell cycle genes when analyzing their scRNA-seq dataset since two out of seven clusters are defined by that signature? I feel that such analysis would be more informative in describing the heterogeneity.

REVIEWERS' COMMENTS

Reviewer #1 (Remarks to the Author):

The authors have thoroughly addressed my concerns. I thank them for their work in clarifying the points raised. I strongly support publication of their work.

We thank the reviewer for his/her time and comments, and for helping us to substantially improve the manuscript throughout the review process with his/her suggestions.

Reviewer #3 (Remarks to the Author):

This is a well done study that describes two subtypes of retinoblastoma. The authors properly addressed the reviewers' comments and revised the manuscript accordingly. I have one minor comment that was not raised in the previous round. Could the authors regress out cell cycle genes when analyzing their scRNA-seq dataset since two out of seven clusters are defined by that signature? I feel that such analysis would be more informative in describing the heterogeneity.

We thank reviewer #3 for taking the time to read the manuscript and assess whether we properly addressed the comments of reviewer #2.

Concerning the comment that was not raised in the previous round:

Before regressing out the cell cycle, we identified five tumor cell clusters corresponding to two phenotypically different populations of tumor cells, cone/neuronal CRX+/EBF3+/GAP43+ tumor cells and cone differentiated CRX+/ARR3+/GUCA1C+ tumor cells. After regressing out the cell cycle scores (G1S score and G2M score by Seurat CellCycleScoring function), at a resolution of 0.5, we identified five clusters of tumor cells corresponding to the same two populations of tumor cells. At a resolution of 0.6, we identified six clusters of tumor cells corresponding to three phenotypically different populations of tumor cells, the same population of cone/neuronal CRX+/EBF3+/GAP43+ tumor cells and now two populations of differentiated cells. The two populations of differentiated cells exhibited distinct features: one population exhibited higher expression of GUCA1C, and was predicted to have a chromosome 10q gain from inferCNV; the other population exhibited lower expression of GUCA1C, but higher expression of ARR3, and was predicted to have both a chromosome 2p and chromosome 10q gain. These results show, in the progression tree proposed in Figure 5e, that the successive copy number alterations of 10q and then 2p are associated with different cone-differentiated phenotypes. After regressing out the cell cycle, there were no clusters corresponding to different cell cycle stages in the differentiated tumor cells. On the contrary, after regressing out the cell cycle, the CRX+/EBF3+/GAP43+ tumor cells were separated into three clusters corresponding to different cell cycle stages (G1, S, G2M) at the two levels of resolutions. Therefore, although we pretreated the expression data by regressing out the cell-cycle scores, cell cycle still plays an important role in clustering for these cells. We are currently testing new methods to remove such cell-cycle effects.

Therefore, as suggested by the reviewer, cell cycle is an important factor that influences the clustering. We prefer not to include our preliminary results after regressing out the cell cycle in the current manuscript, because 1) the main conclusions are the same with or without regressing out the cell cycle and 2) we could not completely regress out the effects of the cell cycle in the undifferentiated tumor-cell population. A finer method to remove such effects needs to be developed. We are planning to develop such methods in a subsequent manuscript, in which we will also study additional cases by single-cell RNA-sequencing.